# Robots are both anthropomorphized and dehumanized when harmed intentionally
Marieke S. Wieringa [1] ✉, Barbara C. N. Müller[2], Gijsbert Bijlstra [2] & Tibor Bosse [2]

The harm-made mind phenomenon implies that witnessing intentional harm towards agents with ambiguous minds, such as robots, leads to augmented mind perception in these agents. We conducted two replications of previous work on this effect and extended it by testing if robots that detect and simulate emotions elicit a stronger harm-made mind effect than robots that do not. Additionally, we explored if someone is perceived as less prosocial when harming a robot compared to treating it kindly. The harm made mind-effect was replicated: participants attributed a higher capacity to experience pain to the robot when it was harmed, compared to when it was not harmed. We did not find evidence that this effect was influenced by the robot's ability to detect and simulate emotions. There were significant but conflicting direct and indirect effects of harm on the perception of mind in the robot: while harm had a positive indirect effect on mind perception in the robot through the perceived capacity for pain, the direct effect of harm on mind perception was negative. This suggests that robots are both anthropomorphized and dehumanized when harmed intentionally. Additionally, the results showed that someone is perceived as less prosocial when harming a robot compared to treating it kindly.

Do we believe robots feel pain? Usually, we do not think of robots as having a "mind" that can experience emotions like happiness, fear, sadness, or pain. Research has however shown that we are automatically more inclined to ascribe a mind to robots when confronted with situations in which they are harmed intentionally[1]. For example, many people expressed their sympathy on social media for hitchBOT, a hitchhiking robot designed as a social experiment on human-robot interaction, who was found beheaded in a park in Philadelphia in 2015[2,3]. This phenomenon, where we suddenly perceive an agent who we usually do not believe experiences pain as a suffering victim, is termed "the harm-made mind"[1,4,5].

As robots become more integrated into society, the harm-made mind phenomenon touches upon the widely debated subject concerning the treatment of robots, specifically social robots, that are designed for interaction with humans[6–11]. For example, is it ethical to allow people to do whatever they want to robots that are designed to elicit a social response? What characteristics of these robots might contribute to the occurrence of a harm-made mind effect? And does the way someone treats a robot influence the way we think about that person? Understanding if specific robot characteristics elicit a strong harm-made mind effect is pivotal to ensure the ethical design of robots, and inform legislation regulating their use. This paper therefore aims to provide a better comprehension of the harm-made mind phenomenon in (social) robots.

One of the robot characteristics that might be particularly important is whether or not it has the ability to detect and express emotions. Social robots are often designed to recognize and express emotions, for example through facial expressions, gestures, and tone of voice[12–14]. Such abilities are deemed quintessential to human-robot communication, as they allow for more natural interactions between robots and humans[15,16]. Research has however shown that people are more likely to attribute moral status to a robot when they believe it is capable of experiencing emotions. For example, attributing affective states to robots decreases the likelihood that we would sacrifice the robot in order to save a human in a moral dilemma[17]. Such findings raise concerns amongst some who worry that emotion-simulating robots are psychologically manipulative[18]. In the current study, we test if robots that recognize and simulate emotions elicit a stronger harm-made mind effect, compared to robots that do not do so. Additionally, we explore whether someone is perceived as less prosocial when they harm a robot, compared to when they treat the robot kindly.

The theoretical rationale behind the harm-made mind comes from the theory of dyadic morality[5,19,20], which postulates that our cognitive template for any moral act (e.g., stealing, killing) consists of a dyad, including a moral agent (e.g., the thief, the killer) who either helps or harms a moral patient (the victim). The theory assumes that morality is strongly linked to mind perception (also called mind attribution), our tendency to ascribe mental

[1]University of Groningen, Campus Fryslân, Leeuwarden, The Netherlands. [2]Behavioural Science Institute, Radboud University, Nijmegen, The Netherlands. ✉e-mail: m.s.wieringa@rug.nl

capacities to others[4]. By constantly ascribing mind to others, we are able to understand their intentions and behavior. Research has found that people usually think about other peoples' minds along two dimensions[21]: the first dimension, *agency*, refers to the capacity to do and intend (e.g., make plans, influence the outcome of situations). The second dimension, *experience*, refers to the capacity for sensation and feeling (e.g., hunger, happiness, sadness).

According to dyadic morality[5,19,20], moral agents are linked to perceptions of agency, specifically to the capacity for intent. Therefore, they can be held accountable for their actions. This link explains why we hold children less responsible for their actions than we do adults, as we tend to ascribe less agency to children than to adults[21,22]. In contrast, moral patients are linked to the perception of experience, in particular to the capacity of experiencing pain. The theory furthermore poses that a moral act *always* requires a complete dyad with both a moral agent and a moral patient. Therefore, when witnessing an interaction in which someone intentionally harms someone else, people are automatically inclined to associate the victim (the moral patient) with the capacity to experience pain, even if they are incapable of doing so. This process, explaining the harm-made mind, is called "dyadic completion".

In line with this theory, ref. 1 found that being part of a moral interaction led to increased mind perception in agents with liminal or non-existent minds. In their vignette experiments, intentional harm towards a robot, a PVS patient, and a deceased person led people to perceive the respective victims to be significantly more capable of experiencing pain. This then led them to attribute more agency and experience to the "mindless" victims generally. Recent studies, using robotic and humanlike avatars[23,24], were able to replicate these findings, though the effect sizes were much smaller compared to ref. 1. There are, of course, multiple factors that could have contributed to these different effect sizes, given, for example, the different stimuli used in these studies. A closer inspection of the vignettes used in ref. 1, however, reveals an interesting design choice. The authors described the robot that was harmed in their study as a complex social robot with "surprising emotional capabilities" such as detecting and expressing several emotions.

We suspect that this ability to detect and simulate emotions amplified the harm-made mind effect, as extant research has shown that the perception of mind in robots is increased by communicative social behaviors[25]. Since the ability to experience emotions is a crucial component of the *experience* dimension of mind perception, strongly emphasizing the ability to detect and express emotions increases the baseline perception of experience in the robot, and thereby the likelihood that participants categorize the robot as a moral patient in the first place. Subsequently, people would be more inclined to believe that it would be wrong to harm the robot[22], and moral wrongness has been found to drive victim perceptions, such as the harm-made mind effect[1,26]. Furthermore, it has been suggested that, when robots evoke an emotional response in the user, the cognitive response that 'it is just a robot' will become less salient[16,27]. In line with these assumptions, previous research has shown that people are less likely to sacrifice a robot in order to save a group of humans when they attribute affective states to the robot compared to when they do not do so[17]. Furthermore, participants are more hesitant to strike a bug-shaped robot with a hammer after reading a story about the robot's personal preferences and experiences, compared to not reading any background story[28].

Recent studies have furthermore found that mind perception in robots appears to be moderated by the presence of socially interactive behavior[25]. For example, one study found that people attributed more mind to a mistreated robot when it has a more detailed articulacy their face (and thus able to simulate humanlike emotional expressions), compared to when the robot has little facial articulacy[16]. We therefore postulate that robots that detect and simulate emotions evoke a stronger harm-made mind effect than robots that do not detect and simulate emotions.

The main focus of this paper is to test if robots that simulate emotions evoke a stronger harm-made mind-effect when they can detect and simulate emotions. There has however additionally been discussion whether

behavior towards robots may causally influence responses towards living beings through a "social robot-human virtue link"[29]. This link poses that cruelty towards robots may foster indifference for cruelty towards other living beings and is based on Kant's argument on animal cruelty: "for a person who already displays such cruelty to animals is also no less hardened towards men"[30]. Following this rationale, witnessing a moral agent intentionally harming a robot might induce the perception that the agent would also behave less prosocial towards animals, and even other people. Research on such a link is however relatively new and empirical evidence scarce. So far, studies that investigated the perception of moral agents have focused on how harmful and benevolent behavior affects perceptions of agency[31–33] One study furthermore reported that watching a video of someone "torturing" a robotic dinosaur caused feelings of anger towards the "torturer" amongst viewers[34], suggesting that moral transgressions against a robot also affect judgments of the moral agent.

In the current research, we present two experiments that partly replicate previous work on the harm-made mind effect by ref. 1 and expand it by investigating if robots that detect and simulate emotions elicit a stronger harm-made mind effect, compared to robots that do not simulate emotions. Based on the theory of dyadic morality and previous research[1,5,23,24], we expect that witnessing intentional harm to a robot will increase the perception that the robot is capable of experiencing pain (H1). We expect that this effect will be stronger when a robot detects and simulates emotions, compared to when it does not (H2). Furthermore, we expect that the perceived capacity for pain will mediate the effect of intentional harm on the attribution of mind more generally (H3). Our main focus will be on the replication and extension of these hypotheses on the harm-made mind effect. Additionally, we test the ancillary hypotheses that someone is perceived as less prosocial when intentionally harming a robot, compared to when they treat the robot kindly (H4). We furthermore explore if this effect is different when a robot detects and simulates emotions, compared to when it does not (H5).

## Methods

We conducted two close replications of ref. 1, as classified based on predefined criteria[35], testing whether witnessing intentional harm towards a robot increased the perception that a robot was capable of experiencing pain. We furthermore expanded this line of work by investigating if the harm-made mind effect in robots was moderated by the ability to detect and simulate emotions. Additionally, we tested if harming a robot influenced perceived prosociality of the moral agent. The research has been independently reviewed by the Ethics Committee Social Sciences (ECSS) of the Radboud University, and there is no formal objection. We preregistered the hypotheses, design, analyses plan, and materials for both experiments on the Open Science Framework (Experiment 1: https://osf.io/mwaub, date of registration 17/08/2020; Experiment 2: https://osf.io/4cezu, date of registration 13/10/2020).

### Participants and design

Both experiments used a 2 (Harm: yes/no) × 2 (Emotions: yes/no) between-subjects design, using the perceived capacity for pain in the robot, general mind perception in the robot, and the perceived prosociality of the moral agent as dependent variables.

For Experiment 1, we made an *a priori* estimate in G*Power[36] for replicating the smallest reported effect size in ref. 1, $d = 0.38$ (which corresponds to $f = 0.19$ for an ANOVA), using an alpha level of .95 and 90% power. This resulted in a required sample size of $n = 294$. Given our available budget, we were able to recruit $n = 452$ participants using the online recruitment platform Prolific. This larger sample size granted us a buffer to account for participants dropping out due to incomplete answers or low quality of data and allowed us to account for the possibility of smaller effect sizes[37], as reported in more recent studies on the harm-made mind effect[23,24]. The experiment was available on all available countries in Prolific, fluent English was set as a pre-screening criterium. We excluded $n = 22$ participants because they filled out the questionnaire in less than half of the median

response time (i.e., less than 134.25 s). Another participant was excluded because their session failed to time-out after the maximum allowed response time of 30 min was exceeded. The final sample consisted of 429 participants (51% men, 48% women, 1% other, mean age = 28 years, $SD$ = 9.33, range 18–66 years). A sensitivity analysis revealed that our final sample allowed to detect effect sizes $f$ = 0.136 for fixed, special, main effects and interaction effects within ANOVA and $f^2$ = 0.026 within linear multiple regression (3 predictors) considering 80% power.

For Experiment 2, we conducted an a priori power analysis based on the effect sizes in Experiment 1. To achieve a minimum of 80% power would require a sample size of 555 participants. In order to account for participants dropping out due to poor data quality (e.g., response time below half the median), we recruited $n$ = 700 via the online recruitment platform Prolific. The experiment was open to all available countries in Prolific, participants were paid £0,63 for their participation. For Experiment 2, we set English as a first language as a pre-screening criterium, as well as the exclusion of participants from Experiment 1. Twenty-one participants were excluded because they filled out the questionnaire in less than half of the median response time (i.e., less than 246 s), as described in the preregistration for this experiment. Additionally, another two participants were dropped because they indicated that they were younger than 18 years of age. The final sample included 677 participants (38% men, 61% women, 1% other, mean age = 34.59, $SD$ = 12.27, range 18–75 years). A sensitivity analysis revealed that our final sample allowed to detect effect sizes $f$ = 0.108 for fixed, special, main effects, and interaction effects within ANOVA and $f^2$ = 0.016 within linear multiple regression (3 predictors) considering 80% power. Table 1 shows an overview of the number of participants per experimental group in each experiment.

### Stimulus material
As we aimed to conduct close replications of ref. 1, we used the exact same textual vignettes to manipulate harm as in the original paper for Experiment 1. We created one new vignette for our Emotions extension. For Experiment 2, we made minor adjustments to the original vignettes that we describe later on. All vignettes can be found in the supplemental material (see Supplementary Table 1a and Supplementary Table 1b) and on OSF.

The vignette used to manipulate Emotions described a robot named "George". This vignette varied depending on the two levels of the factor Emotions (yes/no). For the robot that detects and simulates emotions, we used the vignette previously used by ref. 1. This vignette describes George as a complex social robot that can perform several thousand motor movements and has surprising emotional capabilities, such as the ability to detect and show several emotions (including happiness, surprise, and fear). For the robot that does not simulate emotions, we created a similar description in which George is described as a complex social robot that can perform several thousand motor movements but does not have any emotional abilities. We chose to explicitly emphasize the inability to detect and simulate emotions, rather than just not highlighting this capability, as it is likely that the latter would have resulted in a high variety of perceptions concerning the Emotions of the robot that, in the absence of direct experience, would be based on mental models transferred by the media[38,39].

A pretest ($n$ = 75) showed that participants did perceive the robot that detects and simulates emotions significantly more capable of experiencing various feelings and emotions ($M$ = 3.25, $SD$ = 1.53, range = 1.0–6.0, measured on a 7-point Likert Scale) than the robot without this ability ($M$ = 1.48, $SD$ = 0.97, range = 1.0–4.22), $t$(62.78) = −5.98, $p$ < 0.001, $d$ = 1.38. There was no significant difference between the types of robots in terms of the perceived capacity to think and act (simulating emotions: $M$ = 4.24, $SD$ = 1.32; not simulating emotions: $M$ = 3.9, $SD$ = 1.42), $t$(73) = −1.02, $p$ = 0.310). It must be noted that, at an average of 3.25 on a 7-point scale, participants were not fully convinced that the emotion-simulating robot is actually capable of experiencing emotions. This is unsurprising, as robots are typically perceived to have low levels of experience and moderate levels of agency[21]. The emotion-simulating robot is thus still likely perceived as having more of an ambiguous mind, whereas the robot that does not

simulate emotion is perceived to have a close to non-existent mind. Importantly for our research, the relative difference between these scores differed significantly.

Harm was then manipulated in a second paragraph. In Experiment 1, this paragraph was identical to the one used by ref. 1. In this paragraph, participants read about the robot's caretaker, Dr. Richardson (the moral agent). In the no harm condition, Dr. Richardson frequently re-oils the robot's circuits. In the harm condition, Dr. Richardson frequently abuses the robot by stabbing a scalpel into its sensors because he is envious of the robot's creator. In Experiment 2, we aimed to make the intended position of the robot as the moral patient in the moral dyad more explicit by emphasizing that the intention of the moral agent was to inflict harm on the robot rather than its creator. To do so, the moral agent, Dr. Richardson, was introduced as the robot's creator, who harmed the robot because he was bored of the tedious process that was required for the robot to remain functional. Additionally, as unintentional harm does not have the same effect on mind perception as intentional harm[1], we now emphasized that the moral agent was aware that his actions were harmful (harm condition) or not (no harm condition) to the robot. Finally, in an attempt to make participants less dismissive of the robot's ability to detect and simulate emotion, we made minor changes to the description of the robot to make the text less suggestive (e.g., "the robot *is/is not* sensitive to emotions of others" instead of "the robot *seems/does not seem* sensitive to emotions of others").

### Procedure
Both experiments followed the same procedure. Participants were invited to participate in "a study about a robot" and were told that they would read descriptions about a robot and his caretaker/creator, and subsequently answer questions about their perception of the robot and his caretaker/creator. After they consented to participate in the experiment, participants were randomly assigned to one of the four conditions. The vignettes used to manipulate Emotions and Harm appeared simultaneously on the same page as two subsequent paragraphs. After reading the vignettes, participants filled out a questionnaire measuring our dependent variables. At the end of the questionnaire, participants were able to leave their thoughts about the study in an open text field. After completing the questionnaire, participants were thanked and paid £0,63 for their participation.

### Measures
Unless indicated otherwise, our measures are based on ref. 1.

**Manipulation check.** Participants were asked to indicate how morally right or wrong they considered Dr. Richardson's action on a 7-point Likert scale ranging from 1 "extremely wrong" to 7 "extremely right".

**Perceived capacity for pain.** We asked participants to indicate the extent to which they agreed that the robot is capable of experiencing pain, measured on a 7-point Likert scale ranging from 1 "strongly disagree" to 7 "strongly agree".

**Mind perception.** To measure mind perception, we presented the participants with 18 capacities (e.g., "George can experience fear", "George can make plans") making up the dimensions of mind as identified by Gray and colleagues[21] as a statement. In ref. 1, a selection of items from Gray and colleagues[21] were used to measure mind perception. As it is unclear what the selection of these items was based on, we decided to include all items except for the item measuring the perceived capacity for pain. Participants rated the statements on a 7-point Likert scale ranging from 1 "strongly disagree" to 7 "strongly agree". We performed a principle component analysis (PCA) to asses if separate factors emerged for agency and experience. In Experiment 1, however, three factors emerged from this PCA, which we labeled as "experience", "agency" and "emotion understanding and morality" (see Supplementary Table 2a for full results). We then ran a second PCA which we forced to load on two

**Table 1 | Number of participants per experimental group in each experiment**

| Emotion | Harm | | | |
|---|---|---|---|---|
| | | Yes | No | Total |
| | Experiment 1 | | | |
| | Yes | 109 | 105 | 214 |
| | No | 111 | 104 | 215 |
| Total | | 220 | 209 | 429 |
| | Experiment 2 | | | |
| | Yes | 174 | 166 | 332 |
| | No | 166 | 171 | 345 |
| Total | | 340 | 337 | 677 |

**Table 2 | Effects on perceived moral wrongness**

| | F | df | p | Effect ($\eta_p^2$) | 95% CI effect |
|---|---|---|---|---|---|
| Experiment 1 ($n = 429$) | | | | | |
| Harm | 524.46 | 1, 425 | <0.001 | 0.55 | 0.50, 0.60 |
| Emotions | 0.43 | 1, 425 | 0.51 | 0.001 | <0.001, 0.016 |
| Interaction | 3.09 | 1, 425 | 0.079 | 0.007 | <0.001, 0.031 |
| Experiment 2 ($n = 677$) | | | | | |
| Harm | 651.963 | 1, 673 | <0.001 | 0.49 | 0.44, 0.54 |
| Emotions | 0.460 | 1, 673 | 0.498 | 0.001 | <0.001, 0.01 |
| Interaction | 3.040 | 1, 673 | 0.082 | 0.004 | <0.001, 0.02 |

factors. This time, the factor analysis successfully extracted two factors labeled "agency" (5 items) and "experience" (13 items), see Supplementary Table 2b. In Experiment 2, the PCA also extracted these factors with the exact same items (see Supplementary Table 2c). We created the subscales for agency (Experiment 1: $\alpha = 0.81$, Experiment 2: $\alpha = 0.84$) and experience (Experiment 1: $\alpha = 0.94$, Experiment 2: $\alpha = 0.97$) by averaging the score of these items. The scale for overall mind perception was created by averaging the score on all 18 items (Experiment 1: $\alpha = .94$, Experiment 2: $\alpha = 0.94$). In addition, we followed ref. 1 and measured the perceived consciousness of the robot as a second aspect of mind perception in the robot. In Experiment 1, we used the single item "George is conscious of himself" (1 = strongly disagree, 7 = strongly agree). In Experiment 2, we added a second item "George is conscious of the people and the world around him" and averaged the score of both items ($\alpha = 0.74$). The results for this variable are not discussed in this paper but can be found in the supplemental materials (see Supplementary Fig. 1).

**Perceived prosociality.** For this measure, which was not included in the study by ref. 1, we used an adapted version of a 16 items scale measuring adults' prosociality[40] (e.g., "James tries to help others", "James intensely feels what others feel"). The scale has been widely used in research on prosociality and has reported high reliability in previous studies[41]. Items were rated on a 5-point Likert scale ranging from 1 "strongly disagree" to 5 "strongly agree". The items were averaged to form an index score (Experiment 1: $\alpha = 0.96$; Experiment 2: $\alpha = 0.97$).

### Reporting summary
Further information on research design is available in the Nature Portfolio Reporting Summary linked to this article.

## Results
### Manipulation check
Following ref. 1 we conducted a two-way (Harm × Emotions) between-subjects ANOVA using perceived moral wrongness as the dependent variable as a manipulation check. The data for Experiment 2 violated the assumption of homogeneity as assessed by Levene's test ($p = 0.017$).

The results of the ANOVA's are summarized in Table 2. All reported p-values refer to the two-sided significance level. There was a significant effect of Harm on the perceived moral wrongness of the moral agent's actions in both experiments. In both experiments, participants in the harm condition rated the moral agent's actions as significantly more morally wrong (Experiment 1: $M = 2.33$, $SD = 1.36$, Experiment 2: $M = 2.72$, $SD = 1.31$) than those in the no harm condition (Experiment 1: $M = 5.14$, $SD = 1.18$, Experiment 2: $M = 5.15$, $SD = 1.16$). Neither experiment found statistically significant evidence for a main effect of Emotions on perceived moral wrongness. Finally, the interaction between Harm and Emotions on perceived moral wrongness failed to reach statistical significance in both experiments.

To quantify evidence for the different models, we conducted a Bayesian ANOVA in JASP using the same variables and default settings for priors and Markov chain Monte Carlo settings. A summary of all Bayesian ANOVA's can be found in Table 6. The Bayesian analysis identified the model including only main effects as the best model; the evidence for this model over the null model was decisive in both experiments, $BF_{10} > 100$[42]. Furthermore, when comparing against the best model, we found no evidence in favor of the model that included the interaction effect in either experiment (Experiment 1: $BF_{10} = 0.081$, Experiment 2: $BF_{10} = 0.053$).

Although we did not find statistically significant evidence for the interaction, the means were in the expected direction in both experiments, which is why we decided to proceed with our main analysis.

### Perceived capacity for pain
To test our hypotheses, we employed 2 (Harm: yes/no) × 2 (Emotions: yes/no) between-subjects ANOVA's on the perceived capacity for pain. Figure 1 shows the distributions of the perceived capacity for pain in each condition for each experiment. In order to meet the assumptions for ANOVA mentioned above, we transformed the variables for the perceived capacity to experience pain using a log10 transformation. The transformation did not change the outcome of the results in terms of significance. For interpretation purposes, we report the means and standard deviations from the untransformed variable.

The results of the ANOVA's are summarized in Table 3. The robot's average perceived capacity for pain was well below the scale midpoint in all conditions, indicating that the participants generally disagreed that the robot was capable of experiencing pain. In line with H1, however, there was a significant small main effect of Harm on the perceived capacity for pain in both experiments. As illustrated in Fig. 1, participants were less inclined to strongly disagree that the robot was capable of experiencing pain when the robot was harmed compared to when the robot was not harmed. Emotions had a medium to large sized direct effect on the perceived capacity to experience in both experiments. Participants disagreed less strongly that the robot was capable of experiencing pain when it simulated emotions than when it did not simulate emotions. Crucially, however, we did not find support for H2 since there was no evidence of a significant interaction between Harm and Emotions. An additional Bayesian analysis identified the model including only main effects as the best model; the evidence for this model over the null model was decisive in both experiments, $BF_{10} > 100$[42] (see Table 6). Furthermore, when comparing against the best model, we found no evidence in favor of the model that included the interaction effect in either experiment (Experiment 1: $BF_{10} = 0.095$, Experiment 2: $BF_{10} = 0.238$).

### Mind perception
Following ref. 1, we tested whether the perceived capacity for pain served as a mediator for mind perception more broadly (H3). We ran simple mediation models using Hayes' PROCESS model 4 (5000 samples), as we prespecified to do so in our preregistration in the case our interaction hypothesis (H2) was rejected. We used Harm as independent variable, the perceived capacity to experience pain (untransformed) as the mediator, and mind perception

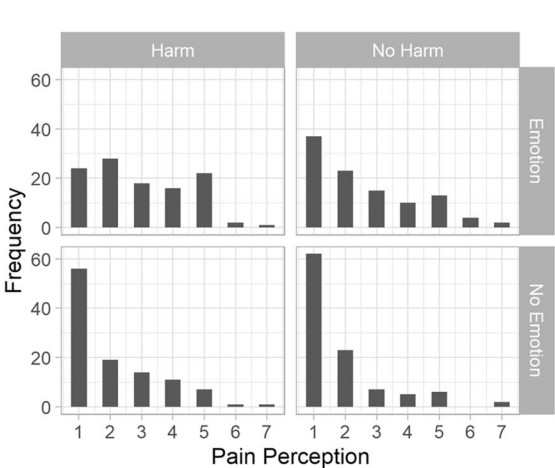

**Fig. 1 | Distributions of pain perception in the robot per condition.** The horizontal axis shows the answer category (7-Point Likert scale, 1 = "strongly disagree the robot can feel pain", 7 = "strongly agree the robot can feel pain") while the vertical axis shows the number of times each category was chosen by our samples. **a** Shows the results for Experiment 1 ($n = 429$), while (**b**) shows the results for Experiment 2 ($n = 677$).

## Table 3 | Effects on perceived capacity for pain

|  | F | df | p | Effect ($\eta_p^2$) | 95% CI effect |
|---|---|---|---|---|---|
| **Experiment 1 ($n = 429$)** |  |  |  |  |  |
| Harm | 6.047 | 1, 425 | 0.014 | 0.01 | 0.0006, 0.044 |
| Emotions | 37.117 | 1, 425 | <0.001 | 0.08 | 0.038, 0.133 |
| Interaction | 0.18 | 1, 425 | 0.67 | <0.001 | <0.001, 0.013 |
| **Experiment 2 ($n = 677$)** |  |  |  |  |  |
| Harm | 14.41 | 1, 673 | <0.001 | 0.02 | 0.005, 0.047 |
| Emotions | 107.19 | 1, 673 | <0.001 | 0.14 | 0.093, 0.185 |
| Interaction | 1.38 | 1, 673 | 0.24 | 0.002 | <0.001, 0.014 |

## Table 4 | Descriptive statistics of variables under investigation

|  | Emotions |  |  |  | No Emotions |  |  |  |
|---|---|---|---|---|---|---|---|---|
|  | Harm |  | No Harm |  | Harm |  | No Harm |  |
|  | *M* | *SD* | *M* | *SD* | *M* | *SD* | *M* | *SD* |
| **Experiment 1** |  |  |  |  |  |  |  |  |
| Pain | 2.95 | 1.54 | 2.61 | 1.66 | 2.09 | 1.42 | 1.84 | 1.35 |
| Mind | 3.76 | 1.17 | 3.82 | 1.11 | 2.51 | 0.98 | 2.51 | 0.92 |
| Agency | 4.13 | 1.42 | 4.37 | 1.27 | 4.13 | 1.38 | 4.09 | 1.17 |
| Experience | 3.55 | 1.34 | 3.61 | 1.29 | 1.89 | 1.03 | 1.90 | 1.06 |
| **Experiment 2** |  |  |  |  |  |  |  |  |
| Pain | 2.73 | 1.82 | 2.14 | 1.45 | 1.57 | 1.24 | 1.34 | 0.97 |
| Mind | 3.39 | 1.28 | 3.58 | 1.21 | 2.17 | 0.92 | 2.00 | 0.73 |
| Agency | 4.00 | 1.31 | 4.20 | 1.26 | 3.69 | 1.59 | 3.53 | 1.38 |
| Experience | 2.39 | 1.47 | 3.35 | 1.47 | 1.58 | 0.95 | 1.42 | 0.75 |

*Mind* Mind perception (overall).

as dependent variable. We included Emotions in the models as a covariate. Table 4 provides the descriptive statistics for all variables under investigation.

The results of the mediation analyses are visualized in Fig. 2 (Experiment 1) and Fig. 3 (Experiment 2). In line with the outcome of the ANOVA's, there was a significant positive effect of Harm on pain perception (a path) in both experiments. In turn, pain perception had a significant positive effect on mind perception in both experiment (b path). In line with H3, there thus was a significant positive indirect effect of Harm on mind perception in both experiments (Experiment 1: $b = 0.13$, 95% CI [0.01, 0.25], Experiment 2: $b = 0.20$, 95% CI [0.10, 0.30]). In contrast to ref. 1, who reported a significant positive total effect of Harm on Mind perception, we did not find statistically significant evidence for a total effect of Harm on Mind perception (the c path) in either of our experiments (Experiment 1: $b = -0.05$, 95% CI [−0.25, 0.15], Experiment 2: $b = -0.01$, 95% CI [−0.17, 0.15]). There was, however, a significant direct negative effect (the c' path) of Harm on Mind Perception in both experiments (Experiment 1: $b = -0.18$, 95% CI [−0.34, −0.02], Experiment 2: $b = -0.21$, 95% CI [−0.33, −0.09], $p < 0.001$). These results suggest a "suppression effect". A suppression effect occurs when the direct and indirect effects of an independent variable have opposite signs, causing the total relationship between X and Y to be insignificant[43-45]. In this case, the negative direct effect of Harm on mind perception was suppressed by the positive indirect effect through the perceived capacity for pain. Finally, Emotions had a positive effect on mind perception, both directly (Experiment 1: $b = 0.91$, 95% CI [0.74, 1.08], Experiment 2: $b = 0.92$, 95% CI [0.79, 1.04], $p < 0.001$) and indirectly (Experiment 1: $b = 0.34$, 95% CI [0.22, 0.47], Experiment 2: $b = 0.48$, 95% CI [0.37, 0.60]).

We repeated the mediation analyses for the separate dimensions of mind perception to explore if perceived agency and experience were affected by harm in different ways. These analyses revealed that harm had a positive indirect effect on perceived agency through the perceived capacity for pain in both experiments (Experiment 1: $b = 0.05$, 95% CI [0.002, 0.11]; Experiment 2: through the perceived capacity for pain), but there was no evidence for a statistically significant direct effect of harm on agency in either experiment (Experiment 1: $b = -0.15$, 95% CI [−0.39, 0.10], $p = 0.25$, Experiment 2: $b = -0.19$, 95% CI [−0.40, 0.02], $p = 0.08$). In sum, we found no statistically significant evidence of a suppression effect for the perceived agency of the robot. Emotions had a significant indirect effect on agency through the perceived capacity for pain in both experiments (Experiment 1: $b = 0.13$, 95% CI [0.05, 0.23], Experiment 2: $b = 0.19$, 95% CI [0.12, 0.28]), but in neither experiment was there statistically significant evidence that

**Fig. 2 | Simple mediation models experiment 1.**
**a** Shows the results of the simple mediation analysis for the effects of Harm and Emotions on overall mind perception in the robot. **b** Shows the effects of Harm and Emotions on the perceived agency of the robot. Finally, (**c**) shows the effects of Harm and Emotions on the perceived experience of the robot. Path coefficients are unstandardized. Asterisks indicate significant paths (*$p < 0.05$, **$p < 0.01$, ***$p < 0.001$). $n = 429$.

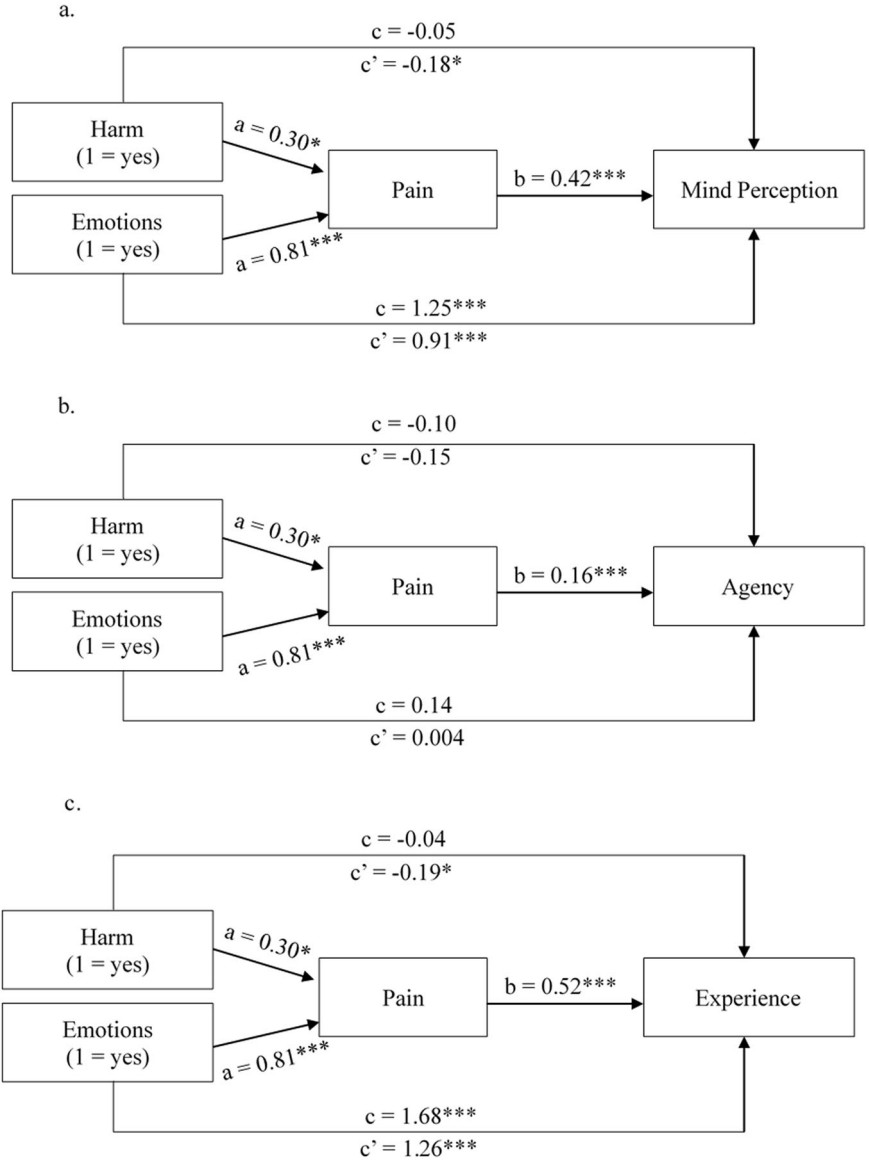

Emotions had a direct effect on agency (Experiment 1: $b = 0.004$, 95% CI [$-0.25$, 0.26], $p = 0.98$]; $b = 0.20$, 95% CI [$-0.02$, 0.42], $p = 0.08$).

For the perceived experience of the robot, harm had a significant positive indirect effect through the perceived capacity for pain in both experiments (Experiment 1: $b = 0.15$, 95% CI [0.009, 0.30], Experiment 2: $b = 0.245$, 95% CI [0.12, 0.38]) and also a significant negative direct effect in both experiments (Experiment 1: $b = -0.19$, 95% CI [$-0.36$, $-0.02$], $p = 0.03$; Experiment 2: $b = -0.22$, 95% CI [$-0.35$, 0.09], $p < 0.001$). Thus, we found evidence of a suppression effect for the perceived experience of the robot. In both experiments, Emotions had a positive effect on perceived experience in the robot both directly (Experiment 1: $b = 1.26$, 95% CI [1.09, 1.44], $p < 0.001$, Experiment 2: $b = 1.19$, 95% CI [1.06, 1.33], $p < 0.001$) and indirectly ($b = 0.42$, 95% CI [0.27, 0.58], Experiment 2: $b = 0.59$, 95% CI [0.45, 0.74]).

**Perceived prosociality of moral agent**
Finally, we tested the ancillary hypotheses that someone is perceived as less prosocial when intentionally harming a robot, compared to when they treat the robot kindly (H4), and if this effect is stronger if a robot simulates emotion (H5). For Experiment 1, the assumption of homogeneity was violated as assessed by Levene's test ($p = 0.001$). Transforming the data did not rectify this problem and so results of the untransformed variable are reported.

We conducted two-way (Harm x Emotions) between-subjects ANOVA's that are summarized in Table 5. The analysis resulted in a significant strong main effect of Harm on the perceived prosociality of the moral agent in both experiments. Participants in the harm condition perceived the moral agent as significantly less prosocial in general than participants in the no harm condition, which provides support for H4. We found no evidence for a main effect of Emotions on the perceived prosociality of the moral agent in Experiment 1. However, in Experiment 2, Emotions surprisingly had a significant small main effect on the perceived prosociality of the moral agent. Participants rated the moral agent slightly higher on prosociality when the robot detected and simulated emotions ($M = 2.95$, $SD = 0.79$) than when it did not ($M = 2.77$, $SD = 0.79$).

Finally, we found no evidence for an interaction effect in Experiment 1. To quantify evidence for the different models, we conducted a Bayesian ANOVA which identified the model including only main effects as the best model; the evidence for this model over the null model was decisive, $BF_{10} > 100$[42] (see Table 6). Furthermore, when comparing against the best model, we found no evidence in favor of the model that included the interaction effect, $BF_{10} = 0.033$. In Experiment 2, however, there was a significant interaction effect between Harm and Emotions on perceived prosociality of the moral agent, but the direction of the effect was different than hypothesized. Post-hoc analyses revealed that there was a significant

**Fig. 3 | Simple mediation models experiment 2.**
**a** Shows the results of the simple mediation analysis for the effects of Harm and Emotions on overall mind perception in the robot. **b** Shows the effects of Harm and Emotions on the perceived agency of the robot. Finally, (**c**) shows the effects of Harm and Emotions on the perceived experience of the robot. Path coefficients are unstandardized. Asterisks indicate significant paths (*$p < 0.05$, **$p < 0.01$, ***$p < 0.001$). $n = 677$.

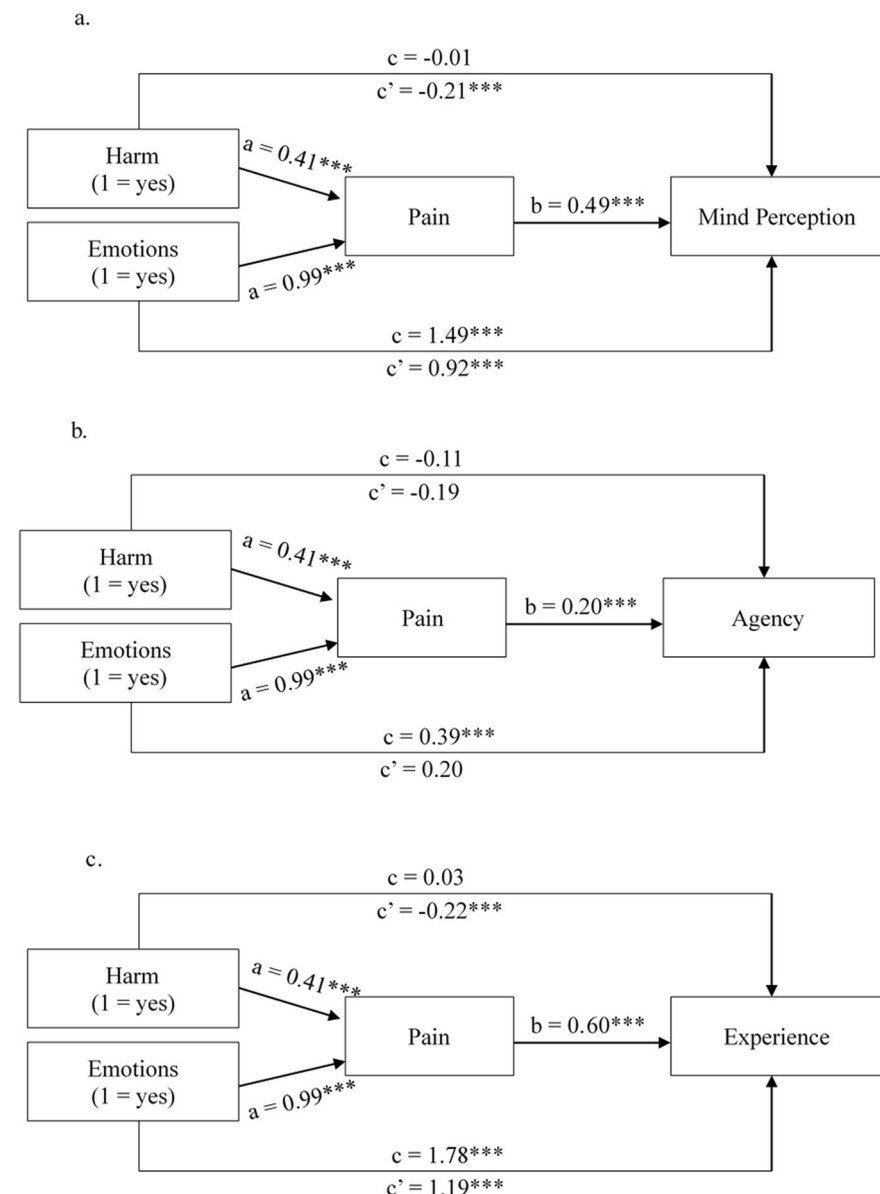

influence of Emotions only when the robot was not harmed. Participants in this condition perceived the moral agent to be slightly more prosocial when the robot detected and simulated emotions ($M = 3.40$, $SD = 0.61$) than when the robot did not do so ($M = 3.06$, $SD = 0.75$), $M_{difference} = 0.34$, $SE_{difference} = 0.08$, $p < 0.001$. Hypothesis 5 is therefore rejected.

## Discussion

The ability for robots to detect and simulate emotions is deemed quintessential to human-robot communication, as it allows for more natural interactions between robots and humans[15,16]. At the same time, however, research suggests that this ability influences both the perception of mind in the robot[25], as well as the perceived moral status of the robot[17,28]. In this paper, we therefore conducted two replications of previous research on the harm-made mind effect in robots[1] and, going beyond previous work, tested if robots that detect and simulate emotions elicit a stronger harm-made mind effect, compared to robots that do not do so. Additionally, we tested the ancillary hypothesis that someone is perceived as less prosocial when harming a robot, compared to treating it kindly.

Across two experiments, we replicated the harm-made mind effect: witnessing intentional harm to a robot increased the perception that

the robot could feel pain. We, however, could not find strong evidence that this effect was moderated by the robot's ability to detect and simulate emotions, nor that harming a robot is deemed more morally wrong when it detects and simulates emotions. These findings were obtained using large online samples, which have been found to produce similar treatment effects as nationally representative population-based samples[46]. Thus, we are fairly confident that our results are generalizable, at least to the western cultures from which the samples have been collected.

The replication of the harm-made mind effect provides support for the theory of dyadic morality[5,19,20], which assumes a strong link between morality and mind perception. Moral agents (e.g., wrong- or gooddoers) are connected to perceptions of agency (e.g., the capacity to intend and act), whereas moral patients (e.g., victims) are connected to perceptions of experience (e.g., sensation and feeling). Our results showed that robots, apparently even ones that explicitly lack the capability to simulate emotions, are perceived as more capable of experiencing pain when they are victimized (i.e., made a victim) by an intentional agent. This supports the idea of dyadic completion, that is, the tendency to perceive a mind in agents with liminal minds when they take part in a moral interaction with an intentional agent.

Our results furthermore show that the harm-made mind effect is quite robust for robots, as we found no evidence that it depends on the ability to detect and simulate emotions. In line with more recent research[23,24], the strength of the harm-made mind effect was significantly smaller than in the seminal study by ref. [1].

Furthermore, we found unexpected conflicting direct and indirect effects of harm on mind perception in the robot. In both experiments, we found a positive indirect effect of harm on mind perception (the harm-made mind effect), as well as a direct negative effect of harm on mind perception in the robot. This is in contrast to ref. [1], who reported a positive direct effect of harm on mind perception in a robot. Exploratory analyses showed that mainly the experience dimension, that is, people's perception concerning the robot's ability for sensation and feelings, was affected negatively by harm. Interestingly, another recent study on the perception of robots that were mistreated found similar conflicting direct and indirect effects of harm on the likeability robots[47].

There are various possible explanations for these conflicting findings. First, the "denial" that others have a mind when they are victimized stands in sharp contrast to the harm made mind, and is typically seen as a form of dehumanization, serving the justification of the inflicted harm[48]. Research has found the denial of human characteristics, such as a mind, is often associated with a diminished moral status[49]. Many people, for example, deny that animals used for consumption have a mind[50,51]. Concerning robots specifically, research has found that the willingness to violate moral principles during human-robot interaction is also grounded in a belief that robots cannot suffer[52]. The negative effect of harm on mind perception could therefore reflect a way to justify the inflicted harm through dehumanization.

Second, these conflicting effects might also reflect automatic and reflexive responses to a robot being harmed[53]. The theory of dyadic morality poses that dyadic completion occurs automatically and implicitly[20], while our automatic responses to robots are also often emotional in nature[54]. At the same time, research has shown that participants see fewer victims in morally impure but ostensibly harmless scenarios when given time to respond than under time pressure[20], indicating that post-hoc rationalization can take place. Given that the participants in our study had ample time to respond and considering the artificial nature of the victim, it is possible that a post-hoc rationalization process conflicted with the automatic process of dyadic completion. Thus, when confronted with the victimization of a robot, participants might find themselves conflicted between an automatic visceral reaction (the harm-made mind effect) and a post-hoc rationalization process (the justification of harm via dehumanization).

Finally, the conflicting findings might be the result of the "uncanny valley of mind", i.e., feelings of eeriness that people experience when confronted with scenario's in which robots recognize and react to emotional states[55]. Research has shown that actively denying mind to robots can serve as a coping mechanism to reduce feelings of uncanniness that these machines can evoke[56]. As our current data does not allow us to draw any conclusions on what causes the conflicting direct and indirect effect of harm on mind perception, an interesting avenue for future research could be to investigate these effects more systematically, for example, by either suppressing or encouraging reflective processes. Additionally, future research could explore how feelings of uncanniness influence the relationship between intentional harm and mind perception in robots.

It is important to point out that our results consistently showed that participants were quite unconvinced that the robot was capable of experiencing much pain, even when the robot was capable of detecting and simulating emotions and when a harm-made mind effect did occur. In the debate concerning the moral status of robots, this is often used as an argument why robots are unworthy of moral consideration[57,58]. Indeed, recent research suggests that the willingness to grant rights to robots amongst the general public depend on the affective and cognitive abilities people believe robots (will) possess[59].

This stance on moral consideration (i.e., basing the moral status of an agent on properties as sentience or consciousness) has however been critiqued by scholars who point out that there is no consensus which exact property defines moral status[10,60]. Additionally, even if there was consensus about the defining property, there would be no way to establish for certain whether or not an agent possesses this quality[61]. Our results indeed showed that the extent to which people believed the robot experiences pain (1) varies across individuals, (2) can be influenced by the characteristics of the robot

**Table 5 | Effects on perceived prosociality of moral agent**

|  | F | df | p | Effect ($\eta_P^2$) | 95% CI effect |
|---|---|---|---|---|---|
| Experiment 1 (n = 429) | | | | | |
| Harm | 237.125 | 1, 425 | <0.001 | 0.36 | 0.29, 0.42 |
| Emotions | 0.244 | 1, 425 | 0.622 | <0.001 | <0.001, 0.014 |
| Interaction | 1.332 | 1, 425 | 0.249 | 0.003 | <0.001, 0.022 |
| Experiment 2 (n = 677) | | | | | |
| Harm | 193.02 | 1, 673 | <0.001 | 0.22 | 0.17, 0.27 |
| Emotions | 11.07 | 1, 673 | 0.001 | 0.02 | 0.003, 0.04 |
| Interaction | 9.30 | 1, 673 | 0.002 | 0.01 | 0.002, 0.036 |

**Table 6 | Results Bayesian ANOVAs**

| Models | $BF_{10}$ | | |
|---|---|---|---|
|  | Moral Wrongness | Capacity For Pain | Perceived Prosociality |
| **Experiment 1** | | | |
| Null model | 1.000 | 1.000 | 1.000 |
| Harm | $2.105_{e+72}$ | 1.731 | $2.451_{e+39}$ |
| Emotions | 0.126 | $3.615_{e+6}$ | 0.111 |
| Harm+Emotions | $2.871_{e+71}$ | $7.227_{e+6}$ | $2.925_{e+38}$ |
| Harm+Emotions+Harm * Emotions | $1.697_{e+71}$ | $1.145_{e+6}$ | $8.202_{e+37}$ |
| **Experiment 2** | | | |
| Null model | 1.000 | 1.000 | 1.000 |
| Harm | $8.326_{e+96}$ | 29.317 | $1.991_{e+34}$ |
| Emotions | 0.100 | $6.083_{e+19}$ | 6.214 |
| Harm+Emotions | $8.622_{e+95}$ | $5.183_{e+21}$ | $3.155_{e+35}$ |
| Harm+Emotions+Harm * Emotions | $4.448_{e+95}$ | $1.232_{e+21}$ | $3.295_{e+36}$ |

Table shows the Bayes Factors ($BF_{10}$) for the different models for each dependent variable. Models are compared to the null model. Analyses were conducted using JASP; default settings were used for priors and Markov chain Monte Carlo settings.

(in this case, the ability to detect and simulate emotions), and (3) differs across contexts (whether or not the robot was harmed intentionally), indicating that mind—and thereby morality—is often in the eyes of the beholder.

Finally, our results consistently showed that someone who harms a robot is perceived as less prosocial than someone who treats a robot kindly. It must be noted that these findings could reflect a general aversion against harmful actions in itself[62]. Nevertheless, the results also showed that providing active care towards a robot that had the ability to simulate emotion resulted in higher perceived prosociality of the robot's creator than when the robot did not have this ability. Therefore, the finding should, in our view, warrant more scientific attention. Currently, most research studying interactions between humans and robots focuses on a single human interacting with a single robot[63]. More research is however needed to examine if interactions with social robots have the potential to influence our perception of other people in a way that is different than our interactions with other objects, since this would carry important implications for people who work with social robots in group settings (e.g., teachers, nurses).

## Limitations

The main limitation of the current studies is that the results are based on text-based vignettes. Using vignettes allowed us to closely replicate previous research[1] and maintain high internal validity. Additionally, text-based vignettes have been successfully used to manipulate robot characteristics in related studies on causes and consequences of mind perception in robots[17,47,64,65]. Moreover, prior studies have already shown that the harm made mind is evoked through textual and visual vignettes, as well as through videos of robot abuse[1,16,23,24]. Nevertheless, more realistic stimulus material like (videos of) actual robots, would have allowed for a more naturalistic manipulation of both simulated emotions as well as intentional harm. Simulated emotions could be manipulated more naturally when using (videos of) actual robots by having the robot either respond emotionally before, during, and after the harm, or not. Furthermore, using (videos of) robots as stimulus material would rule out the possibility that the results are influenced by specific wordings (i.e., stabbing and twisting a scalpel in the robot's sensor vs. kicking the robot's surface). One could argue, for example, that especially the mention of damaged sensors in the harm conditions of the current stimulus material might have interfered with the emotion manipulation, thereby possibly causing the absence of an interaction effect. There would also not be a need to specify the moral agent's motivation behind the harm when using a video.

Using textual vignettes as stimulus material, we were not able to find evidence that the harm-made mind effect is moderated by the robot's ability to detect and simulate emotions. However, prior studies did find evidence that mind perception in robots is moderated by (humanlike) characteristics of the robot[16,25]. Additionally, people perceive robots more positively when they are physically present[66], and empathize more with physically present robots than mediated ones[67]. Therefore, it is possible that a more naturalistic manipulation of simulated emotions might still reveal a significant role of simulated emotions in the harm-made mind effect. Finally, using actual human-robot interaction also provides the opportunity to include physiological and behavioral measures that help to better capture our oftentimes automatic responses to robots.

Our analyses were furthermore conducted solely on a group-level. Research has however shown that there are individual differences in both the tendency to perceive mind in nonhuman agents[68,69] as well as in how people respond to robots[70,71]. It has been suggested that, in such cases, group-level findings may not always translate to the person-level (the group-to-person generalizability problem[72]). An interesting avenue for future research could therefore be to explore the harm-made mind effect on the person-level through within-subjects designs.

Finally, future research should consider a control condition where the agent simply does nothing to the robot. In the current studies, the control (no harm) condition described how the moral agent would frequently re-oil

the robots circuits. Prior research has however found that actively caring for the robot by upholding its maintenance is regarded by some as a way to treat the robot with moral consideration[73]. Following the theory of dyadic morality, research has also found that caring for a robot by repairing it when it breaks (compared to moving it to an adjacent room and doing nothing) increased mind perception through the perceived capacity to experience joy, though only when the participant took on the perspective of the moral agent and not when taking on the perspective of a bystander[74]. Given that participants in this study were placed in a bystander perspective, it is unlikely that such an effect occurred for our control condition.

## Conclusion

Witnessing intentional harm to a robot increases the perception that the robot can feel pain and, in turn, the perception that it has a mind. We did not find convincing evidence that this effect is dependent on the robot's ability to detect and simulate emotions. At the same time, however, harm also negatively affects mind perception. These results suggests that robots might be simultaneously humanized and dehumanized when harmed intentionally. Finally, our results consistently show that someone who intentionally harms a robot is perceived as less prosocial, compared to someone who treats the robot kindly.

## Data availability

All data collected for this publication are publicly available on the Open Science Framework (https://osf.io/b4nqc/).

## Code availability

The codes that were used to run the analyses for this publication are publicly available on the Open Science Framework (https://osf.io/b4nqc/).

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

## Author contributions

Marieke S. Wieringa: Conceptualization, Methodology, Software, Validation, Formal Analysis, Investigation, Resources, Data Curation, Writing—Original Draft, Project Administration. Barbara C.N. Müller: Conceptualization, Methodology, Resources, Data Curation, Writing—Review & Editing, Supervision. Tibor Bosse: Conceptualization, Methodology, Resources, Validation, Data Curation, Writing—Review & Editing, Supervision. Gijsbert Bijlstra: Data Curation, Writing—Review & Editing, Supervision.

## Competing interests

The authors declare no competing interests.
