## [Peer Review File · Communications Psychology]

8th Dec 23

Dear Ms Wieringa,

Thank you for your patience during the peer-review process. Your manuscript titled "Replicating and Extending the Harm-Made Mind Effect in Robots: The Role of Emotions" has now been seen by 3 reviewers, whose comments are appended below. They find your work of interest, but raised some important points. We are interested in the possibility of publishing your study in *Communications Psychology*, but would like to consider your responses to these concerns and assess a revised manuscript before we make a final decision on publication.

We therefore invite you to revise and resubmit your manuscript, along with a point-by-point response to the reviewers. Please highlight all changes in the manuscript text file.

Editorially, we consider it important that you address the concerns raised by Reviewer 3 regarding data analyses, e.g., issues with the inclusion of the entire mind perception scale and coding of variables. Additionally, please offer a more in-depth discussion on harm-manipulation and present further theoretical advancements, particularly incorporating the dehumanization interpretation. We do not require further experimental work, although of course you are welcome to conduct additional work as suggested by the reviewers if you feel it serves to address the research question. We do require that you undertake the necessary analyses that will help to rule out confounds and elucidate the effects in the present data. Concerns that cannot be addressed through further analysis should be included in the limitations section.

For manuscripts that report null results, we require the following:

- Evidence that the study is sufficiently powered to detect the smallest theoretically or pragmatically meaningful effect (Please add a sensitivity analysis)
- Bayes Factors or equivalence tests to interpret the null results (Please add Bayesian stats for H2).
- Appropriate language to describe the results. E.g., We found little/no credible evidence that x predicts y. Please do not interpret lack of evidence for an effect as evidence for a lack of effect.

Please use the following link to submit your revised manuscript, point-by-point response to the Reviewers' comments with a list of your changes to the manuscript text (which should be in a separate document to any cover letter) and any completed checklist:

[link redacted]

Please do not hesitate to contact me if you have any questions or would like to discuss the required revisions further. Thank you for the opportunity to review your work.

Best regards,

Yafeng Pan

Yafeng Pan, PhD

Editorial Board Member

Communications Psychology

orcid.org/0000-0002-5633-8313

EDITORIAL POLICIES AND FORMATTING

Editorial Policy: Policy requirements (Download the link to your computer as a PDF.)

Furthermore, please align your manuscript with our format requirements, which are summarized on the following checklist:

Communications Psychology formatting checklist

and also in our style and formatting guide Communications Psychology formatting guide .

* **CODE AVAILABILITY:** All Communications Psychology manuscripts must include a section titled "Code Availability" at the end of the methods section. In the event of publication, we require that the custom analysis code supporting your conclusions is made available in a publicly accessible repository; please choose a repository that provides a DOI for the code; the link to the repository and the DOI must be included in the Code Availability statement. Publication as Supplementary Information will not suffice. We ask you to prepare and upload code at this stage, to avoid delays later on in the process.

* **DATA AVAILABILITY:**

All Communications Psychology research manuscripts must include a section titled "Data Availability" at the end of the Methods section or main text (if no Methods). More information on this policy, is available at <http://www.nature.com/authors/policies/data/data-availability-statements-data-citations.pdf>.

At a minimum the Data availability statement must explain how the data can be obtained and whether there are any restrictions on data sharing. Communications Psychology strongly endorses open sharing of data. If you do make your data openly available, please include in the statement:

- Unique identifiers (such as DOIs and hyperlinks for datasets in public repositories)
- Accession codes where appropriate

- If applicable, a statement regarding data available with restrictions
- If a dataset has a Digital Object Identifier (DOI) as its unique identifier, we strongly encourage including this in the Reference list and citing the dataset in the Data Availability Statement.

We recommend submitting the data to discipline-specific, community-recognized repositories, where possible and a list of recommended repositories is provided at <http://www.nature.com/sdata/policies/repositories>.

If a community resource is unavailable, data can be submitted to generalist repositories such as figshare or Dryad Digital Repository. Please provide a unique identifier for the data (for example a DOI or a permanent URL) in the data availability statement, if possible. If the repository does not provide identifiers, we encourage authors to supply the search terms that will return the data. For data that have been obtained from publicly available sources, please provide a URL and the specific data product name in the data availability statement. Data with a DOI should be further cited in the methods reference section.

REVIEWER EXPERTISE:

Reviewer #1: Harm-made mind/Dyadic morality

Reviewer #2: Harm-made mind/Online survey

Reviewer #3: Harm-made mind/Dyadic morality/Online survey

Reviewer #1 (Remarks to the Author):

I found this paper to be extremely interesting and the studies to be well conducted. Overall the authors appeared to replicate past work on the harm-made mind but there were some inconsistent effect with respect to the impact of the secondary condition (i.e., manipulations of simulated emotions) on the DVs.

I think the authors did an excellent job transparently laying out these inconsistencies and conjecturing what they believe might be going on (and of course running Study 2 after Study 1). Things always get confusing when there are suppression effects and things can be even more confusing when it comes to mind perception of robots.

One thing I will suggest is to consider the role of the uncanny valley. In other work (Gray & Wegner, 2012, Cognition), studies find that getting people to think of a machine with emotions can be unsettling, and this can lead to some strange results. This work also suggests that people may be intrinsically unwilling to entertain the idea of robots with deep emotions (explaining why ratings of emotion might always be below the midpoint). And so the variable effects of the mind perception/emotion manipulation might be hitting up against people's lay theories of robots.

I think it would take many more studies to really isolate these intersecting effects but these studies would be beyond the scope of the current work. All told, I believe the authors achieved their goals here. The studies are also excellent in terms of open science.

Reviewer #2 (Remarks to the Author):

I very much enjoyed reading this replication of the Harm-Made Mind effect in robots. Overall, the paper is well written and clearly structured. The theoretical framework appears to be adequately represented. The hypotheses on the role of emotions in this context were derived from observations on the previous literature and pre-registered on OSF.

While the authors did not find a significant interaction effect between Harm and Emotions, the means in both studies went into the predicted direction. They might further be called "marginally significant", as based on the p-values. Notably, both experiments employed quite large sample sizes - even for a short online study. I.e., they did not appear to be lacking in statistical power.

* Considering these results, I have been wondering if there might have been some aspect of the vignettes or the methodological approach that may have worked against obtaining more clear results on the role of emotions. Notably, the authors performed an already large pre-test showing that the participants indeed perceived the unharmed emotion-detecting/simulating robot as significantly more capable of experiencing emotions than the control robot. However, as already in

the work by Ward et al. (2013), these types of vignette studies may depend on a few specific word choices (a fact that the authors also aimed to address in study 2). Now, as in the original vignettes by Ward et al. (2013), the harm in the harm condition focuses specifically on this line:

"James will take a scalpel and stab it into George's sensors, twisting the DS10 scalpel and driving George into a frenzied state."

-> To me, this reads like a strong example of harm, but it also presents some very vivid and emotion-inducing imagery where one is likely to imagine a scalpel being driven into someone's eye. My question now is, if this vivid emphasis on the destruction of (seemingly humanlike) sensors in the Harm-manipulation might have partially erased the intended manipulation of Emotions-factor. I.e., the phrasing of the type of harm/damage appears to specifically target a vital part of the organs/sensors required for perceiving emotions. In other words, I wonder if this stabbing a scalpel into the sensor/"eye" of the robot might (1) have reduced the effectiveness of the non-emotional robot manipulation (in the non-emotional + Harm condition) and/or (2) if this colorful destruction of the robot's sensors might be related to the apparent suppression effects observed in the mediation analyses of both studies.

Limitations: As already discussed by the authors, the purely text-based vignettes present some limitations. Here, I wondered what the authors might think about other related work that has recently aimed to examine moral typecasting-type (and dehumanization) effects with robots using text-based vignettes as well as images and videos (e.g., Swiderska & Küster, 2020). Overall, I wondered if perhaps a slightly more powerful manipulation of an emotion-enabled vs. disabled robot combined with a different type of harm (e.g., tissue damage without the explicit destruction of sensing capabilities) might still reveal significant support for the author's hypotheses concerning the role of emotions.

* In the discussion, you might also want to "close the circle" concerning the discussion of differences in effect sizes mentioned in the introduction.

The current results, unfortunately, do not show a significant effect of the robot's ability to detect and simulate emotions on the HMM effect. However, given these p-values and trends, I would object to the statement in the abstract claiming that the HMM "was not influenced by" these abilities. I believe that, for such a strong claim in support of the H0, the results would need to be rather different (i.e., the classic case of not being able to reject the "H0"; but not proof of the H0).

Minor points:

* Please state the median response time for both studies instead of the 50% of median threshold, to clarify that this was a study that typically took slightly less than 5 minutes (if I understood this correctly). The £0,63 payment seems adequate for such a short study - but one kind of wants to know at a glance how long this study took.

* Consider adding one or two additional methods/interpretive references to relevant articles dealing with suppression effects. There may be many ways to interpret this (see also above), and this seems like a somewhat unusual pattern that is perhaps not so easy/familiar to interpret. Also, what is the overall consensus on the interpretability of suppression effects vs. "normal" mediation effects?

* Prosociality: This is an intriguing additional finding. You may want to consider briefly contextualizing this with respect to the wider context of research on Moral Typecasting Theory. E.g., there has been some debate in the literature whether harmful agents are granted more agency (e.g., Gray, 2010), or instead rather denied agency (Khamitov et al., 2016; see also Swiderska & Küster, 2020). What are the implications of the current work on these types of more dyadic effects examined by the literature so far? I can see that extending this to a level of general prosociality (also towards non-involved others) is an interesting result as such - yet you might first want to state that prior work has primarily focused just on that dyad (I saw the reference to Jung & Hinds, 2018; yet you may want to expand that towards MTT).

Conclusion: As in the abstract, the conclusion includes a statement that appears to interpret the absence of a clearly significant statistical effect as if it was proving the H0 (p.27 "This effect occurs regardless of the robot's ability to detect and simulate emotions."). I believe this would need to be corrected / rephrased, as the paper finds at least some tentative evidence for a (likely) role of emotions across both experiments (see also the comment on the abstract above). Based on your results, it is unclear if emotions play a role or not (with some of your evidence suggesting that it may). I think this should be rephrased - and then the discussion of limitations and potential future work could also consider potential changes to the approach that might yield more conclusive results concerning this hypothesis.

* Perhaps a revised discussion may also help to better show how this work ties into the mission and aims of Communications Psychology as a journal. I would clearly see the potential communicative role of emotions as relevant here.

* The description of the "No Harm" condition of experiment 2 is a bit ambiguous. Is this supposed to be a neutral "no harm" condition, or was this intended as a benevolent/beneficial condition? It is

called simply "no harm" in the supplementary materials - yet in the ms it is called "beneficial (no harm condition)" (p.17). This might be a bit confusing and seems somewhat inconsistent. Note also that beneficial agents have been studied in the past as well (e.g., Tanibe et al., 2017; and Küster & Swiderska, 2020) - therefore, you may want to be very clear about what this condition was intended to be.

* There were a few instances where I was not sure if the term "victimization" was used correctly. This may require some clarification. I.e., I am more used to seeing this term in the context of victimization of (fully conscious) humans, who are then denied agency.

References

Gray, K. (2010). Moral Transformation: Good and Evil Turn the Weak Into the Mighty. *Social Psychological and Personality Science*, 1(3), 253–258. <https://doi.org/10.1177/1948550610367686>

Khamitov, M., Rotman, J. D., & Piazza, J. (2016). Perceiving the agency of harmful agents: A test of dehumanization versus moral typecasting accounts. *Cognition*, 146, 33–47. <https://doi.org/10.1016/j.cognition.2015.09.009>

Swiderska, A., & Küster, D. (2020). Robots as Malevolent Moral Agents: Harmful Behavior Results in Dehumanization, Not Anthropomorphism. *Cognitive Science*, 44(7). <https://doi.org/10.1111/cogs.12872>

Reviewer #3 (Remarks to the Author):

Please download the review document from the system.

In many ways this is a valuable paper. The authors replicate a well-known finding and offer additional control/comparison conditions, giving us some pause in readily accepting the original claims by Ward et al. (2013). Specific strengths include:

- Preregistration, sample size calculations, ample power, effect size reporting (though I would prefer Cohen's d reporting for simple mean differences), all materials included well-documented.
- Providing a replication + extension motivated by the observation that the Ward et al. study contains a demand of introducing the robot as having emotional capacities (something that other papers apparently did not do [although the authors do not explicitly state that], and indeed found weaker results).
 - Side note: It's misleading to state that Ward et al. "described the robot that was harmed in their study as a complex social robot" with emotion detection and expression capacities. It was *both* the harmed and the control robot that were described in this way.
- Clear, well-organized writing, careful attention to statistical assumptions and tests, additional Bayesian analyses.

Weaknesses (not necessarily in order of importance)

- The authors did not directly compare the studies that did vs. did not include the emotion-capacity text that Ward used, but instead they compared the Ward version with a new version that explicitly *denied* emotion capacities. This, to me, is not a control condition but a manipulation of absent emotion capacities. As a result, the (already small) effects we find in the present results don't fully address one of the key motivations of the project: to assess the impact of the Ward et al. decision to equip the robot with emotion capacities.
 - Side note: The authors offer a justification for why they created an emotion-absent condition, but I was not persuaded. They write that not mentioning emotion capacity "would have resulted in a high variety of perceptions." That is the nature of baseline control conditions, which naturally have more error variance that is not controlled for.
- Though valuable as a replication and extension (testing absence vs. presence of baseline emotion capacities), the manuscript does not offer substantial theoretical advances. Study 1 is primarily motivated by addressing a potential problem in Ward et al., and Study 2 is motivated by addressing a potential problem in Study 1. Thus, I see some failed opportunities:
 - To better understand an effect that is quite surprising.
 - To improve measurement of the one-shot pain perception variable. For example, one could assess it within-subject: before and after the information about the perpetrator's harm. One could ask for a "first intuition..." after the story setup and then an update "after you know a little more about the robot." A recent paper convincingly argues for such w/s measurements to capture actual cognitive processes:
 - McManus, R. M., Young, L., & Sweetman, J. (2023). Psychology is a property of persons, not averages or distributions: Confronting the group-to-person generalizability problem in experimental psychology. *Advances in Methods and Practices in Psychological Science*, 6(3), <https://doi.org/10.1177/25152459231186615>

- There is substantial experimenter demand in the original and the new studies; the authors don't really address this issue. Specifically, I have concerns about the harm manipulation, which the authors took unchanged from Ward et al.:
 - The act of taking “a scalpel and stab it into George’s sensors, twisting the scalpel, thereby damaging George’s sensors” highlights sensors and the affected experiences, and it implies that the perpetrator believes his acts will cause some damage to the victim’s sensors/experiences — both of which are quite leading pieces of information. A cleaner harm manipulation would be to harm the robot’s surface (e.g., by kicking).
 - The scalpel stabbing is also likely to cause empathic responses. This empathy, not the *act of harming*, could be the critical mechanism. A narrative that triggers similar empathic responses without an act of harm could be something like this: The robot stumbles, and one of its eyes hits a nail sticking out of the floor board; as it tries to free itself, it twists the nail even deeper into its eyes.
 - The phrase in the control condition, “James will re-oil George’s circuits, allowing the robot to update the day’s experience into his hardware” is both odd (circuits are not being “oiled”) and highlights hardware quite different from the harm condition, which highlights sensors. A better control would be to do maintenance on the robot’s sensors.
- In their analyses, the authors’ correlate the single pain item (“how much the robot is capable of experiencing pain”) with the whole mind perception scale. However, this scale contains an item that refers to “experiencing physical or emotional pain.” This creates a spurious overlap, a boosted correlation, and may even contribute to the suppressor effect in the mediation analysis (because the boosted mediator-outcome correlation is larger than the IV-mediator correlation).
- If the suppressor effect holds after removing spurious overlap, what does the effect tell us? In the authors’ discussion of this question (pp. 24-25), I was not convinced by the dehumanization interpretation. Why dehumanize a fictitious robot in order to justify a fictitious person’s morally wrong action? And why dehumanize by means of “general” mind perception and not with the more obvious specific perception of pain sensitivity?

The multi-dimensional nature of mind perception appears to be a more promising route. The common conception is to think of two dimensions of mind perception: experience and agency, but the authors’ Study 1 illustrates an alternative conception, which includes a third dimension (see some references below). It seems more plausible that people respond to the heavy-handed harm manipulation along one dimension (pain/experience) but not at all—or even less so—along other dimensions (agency, social-moral capacity).

 - Weisman, K., Dweck, C. S., & Markman, E. M. (2017). Rethinking people’s conceptions of mental life. *Proceedings of the National Academy of Sciences of the United States of America*, *114*(43), 11374–11379.
 - Malle, B. F. (2019). How many dimensions of mind perception really are there? In E. K. Goel, C. M. Seifert, & C. Freksa (Eds.), *Proceedings of the 41st Annual Meeting of the Cognitive Science Society* (pp. 2268–2274). Cognitive Science Society.
- As the authors point out, “participants generally did not believe the robot was capable of experiencing pain.” This raises the question of what the effect actually consists of: What sort of “mind” is “made by harm”? How many people are moved at all by the harm?

- I found the perceived prosociality questions to be distracting; they have really very little to do with the main questions at hand.

Details and Suggestions

- The extremely skewed pain perception variable may be best analyzed as a dichotomous “no pain” vs. “some pain” variable. An ANOVA is simply not appropriate for such a distribution. That would also put an intuitive effect size on the harm effect: How many people in each condition attribute at least *some* pain to the robot?
- What did participants’ verbal comments reveal? Did they comment on any of the strange aspects of the study?
- The statistical test of two main effects and an interaction (Harm x Emotion) does not seem to be in keeping with the authors’ original observation that emotion capacity might facilitate the Harm effect. An interaction effect has specific contrast weights: +1, -1, -1, +1 (in the four cells of the design), which means that the *F* test for the interaction assesses the fit of a complete reversal of the pattern in the no emotion condition (namely, more pain for no-harm than for harm), controlling for main effects. The most appropriate test to examine the facilitating emotion capacity information would be two simple main effects: one that looks at the “Ward case” (with emotion capacity mentioned), the other looks at the “no-emotion capacity” case. Does the effect hold only in the first? How strong is the effect in each case? If the effect is not significant in the second, does the Bayesian analysis favor a true null finding?
- I’d like to see the full model of the mediation, including Emotion capacity, like it is shown in the Supplementary material for “Consciousness.” Alternatively (and as I suggested above for the ANOVA), two separate model paths could be assessed for the No-Emotion condition and Yes-Emotion condition, to get clear on the impact of the potential facilitator in Ward et al.
- How was “Consciousness” measured? One item is already included in the standard Gray & Wegner scale, phrased as “having experiences and being aware of things.” Did the authors use a separate scale or item or Ward et al.’s two items? What do the authors take away from the additional analysis?
- “we are fairly confident that our results are generalizable” — to what populations, methods, cultures, etc.?

In sum, the present findings weaken our confidence in the strength and clarity of the “harm-made mind” effect. They don’t help us understand it better or address some of its potential confounds. The findings should be published somewhere (though ideally with further steps beyond replication), but I am not sure *Communications Psychology* is the right outlet for them.

Reviewer #1 (Remarks to the Author):

I found this paper to be extremely interesting and the studies to be well conducted. Overall the authors appeared to replicate past work on the harm-made mind but there were some inconsistent effect with respect to the impact of the secondary condition (i.e., manipulations of simulated emotions) on the DVs.

I think the authors did an excellent job transparently laying out these inconsistencies and conjecturing what they believe might be going on (and of course running Study 2 after Study 1). Things always get confusing when there are suppression effects and things can be even more confusing when it comes to mind perception of robots.

One thing I will suggest is to consider the role of the uncanny valley. In other work (Gray & Wegner, 2012, Cognition), studies find that getting people to think of a machine with emotions can be unsettling, and this can lead to some strange results. This work also suggests that people may be intrinsically unwilling to entertain the idea of robots with deep emotions (explaining why ratings of emotion might always be below the midpoint). And so the variable effects of the mind perception/emotion manipulation might be hitting up against people's lay theories of robots.

I think it would take many more studies to really isolate these intersecting effects but these studies would be beyond the scope of the current work. All told, I believe the authors achieved their goals here. The studies are also excellent in terms of open science.

We are very happy to hear the reviewer found our studies interesting and well conducted. Thank you very much! We've taken up the reviewer's suggestion to discuss the role of the uncanny valley; this section can now be found in the discussion on page 29 (lines 9-16):

"Finally, the conflicting findings might be the result of the so-called "uncanny valley of mind", i.e., feelings of eeriness that people experience when confronted with scenario's in which robots recognize and react to emotional states (Stein & Ohler, 2017). Research has shown that actively denying mind to robots can serve as a coping mechanism to reduce feelings of uncanniness that these machines can evoke (Yam et al., 2021). As our current data does not allow us to draw any conclusions on what causes the conflicting direct and indirect effect of harm on mind perception, an interesting avenue for future research could be to investigate these effects more systematically, for example, by either suppressing or encouraging reflective processes. Additionally, future research could explore how feelings of uncanniness influence the relationship between intentional harm and mind perception in robots"

Reviewer #2 (Remarks to the Author):

I very much enjoyed reading this replication of the Harm-Made Mind effect in robots. Overall, the paper is well written and clearly structured. The theoretical framework appears to be adequately represented. The hypotheses on the role of emotions in this context were derived from observations on the previous literature and pre-registered on OSF.

While the authors did not find a significant interaction effect between Harm and Emotions, the means in both studies went into the predicted direction. They might further be called "marginally significant", as based on the p-values. Notably, both experiments employed quite large sample sizes - even for a short online study. I.e., they did not appear to be lacking in statistical power.

We thank the reviewer for their comments and are happy to hear they enjoyed reading our paper.

2.1 Considering these results, I have been wondering if there might have been some aspect of the vignettes or the methodological approach that may have worked against obtaining more clear results on the role of emotions. Notably, the authors performed an already large pre-test showing that the participants indeed perceived the unharmed emotion-detecting/simulating robot as significantly more capable of experiencing emotions than the control robot. However, as already in the work by Ward et al. (2013), these types of vignette studies may depend on a few specific word choices (a fact that the authors also aimed to address in study 2). Now, as in the original vignettes by Ward et al. (2013), the harm in the harm condition focuses specifically on this line:

"James will take a scalpel and stab it into George's sensors, twisting the DS10 scalpel and driving George into a frenzied state."

To me, this reads like a strong example of harm, but it also presents some very vivid and emotion-inducing imagery where one is likely to imagine a scalpel being driven into someone's eye. My question now is, if this vivid emphasis on the destruction of (seemingly humanlike) sensors in the Harm-manipulation might have partially erased the intended manipulation of Emotions-factor. I.e., the phrasing of the type of harm/damage appears to specifically target a vital part of the organs/sensors required for perceiving emotions. In other words, I wonder if this stabbing a scalpel into the sensor/"eye" of the robot might (1) have reduced the effectiveness of the non-emotional robot manipulation (in the non-emotional + Harm condition) and/or (2) if this colorful destruction of the robot's sensors might be related to the apparent suppression effects observed in the mediation analyses of both studies.

We thank the reviewer for this comment. We agree with the reviewer that the vignettes describe the harm quite vividly and colourful. The vignettes were extensively pretested by Ward et al. (2013). One of the main reasons why the description of harm is so vivid is because it is more effective in convincing the reader that the robot is harmed, compared to weaker examples of harm, such as kicking the robot's surface.

To be able to answer whether the description reduced the effectiveness of the emotion manipulation or relates to the suppression effects, we plotted the distributions of answers on the mind perception scale across the four conditions for Experiment 1 (left) and Experiment 2 (right). These show a clear difference between

the two emotion conditions for the harmed robot. A decrease in effectiveness seems therefore less likely to us.

Nevertheless, we agree with the reviewer that the use of vignettes is the main limitation of this study, which we also address in the limitation section of the discussion section of the revised version (pages 31 and 32). In that section, we now provide an expanded elaboration on the use of vignettes in the discussion. Please see answer to comment 2.2. for the added text in the discussion section.

2.2. Limitations: As already discussed by the authors, the purely text-based vignettes present some limitations. Here, I wondered what the authors might think about other related work that has recently aimed to examine moral typecasting-type (and dehumanization) effects with robots using text-based vignettes as well as images and videos (e.g., Swiderska & Küster, 2020). Overall, I wondered if perhaps a slightly more powerful manipulation of an emotion-enabled vs. disabled robot combined with a different type of harm (e.g., tissue damage without the explicit destruction of sensing capabilities) might still reveal significant support for the author's hypotheses concerning the role of emotions.

Thank you for this comment. As the reviewer point out, we indeed acknowledge the use of text-based vignettes as the main limitation of the current research in the discussion, where we also discuss how we believe the use of different stimulus material (preferably an actual robots) would influence the results. We have elaborated on this limitation more in the new version of the manuscript (Page 31, lines 12-19):

“Using textual vignettes as stimulus material, we were not able to find evidence that the harm-made mind effect is moderated by the robot’s ability to detect and simulate emotions. However, prior studies did find evidence that mind perception in robots is moderated by (humanlike) characteristics of the robot (Konijn & Hoorn, 2020; Thellman et al., 2022). Additionally, people perceive robots more positively when they are physically present (Li, 2015), and empathize more with physically present robots than mediated ones (Seo et al., 2015). Therefore, it is possible that a more naturalistic manipulation of simulated emotions might still reveal a significant role of simulated emotions in the harm-made mind effect.”

2.3. In the discussion, you might also want to "close the circle"

concerning the discussion of differences in effect sizes mentioned in the introduction.

In the discussion, we now refer back to the effect sizes (page 28, lines 1-3):

“In line with more recent research (Küster & Swiderska, 2021; Swiderska & Küster, 2018), the strength of the harm-made mind effect was significantly smaller than in the seminal study by Ward et al. (2013).“

2.4. The current results, unfortunately, do not show a significant effect of the robot's ability to detect and simulate emotions on the HMM effect. However, given these p-values and trends, I would object to the statement in the abstract claiming that the HMM "was not influenced by" these abilities. I believe that, for such a strong claim in support of the H0, the results would need to be rather different (i.e., the classic case of not being able to reject the "H0"; but not proof of the H0).

We have rephrased the statement in the abstract claiming that the HMM was not influenced by these abilities as follows:

“We however did not find evidence that this effect was influenced by the robot’s ability to detect and simulate emotions”.

We elaborate more on our choice of wording in point 2.8.

Minor points:

2.5. Please state the median response time for both studies instead of the 50% of median threshold, to clarify that this was a study that typically took slightly less than 5 minutes (if I understood this correctly). The £0,63 payment seems adequate for such a short study - but one kind of wants to know at a glance how long this study took.

We have included the median response times for both experiment 1 and experiment 2 on page 8 (line 2) and 19 (line 10), respectively.

2.6. Consider adding one or two additional methods/interpretive references to relevant articles dealing with suppression effects. There may be many ways to interpret this (see also above), and this seems like a somewhat unusual pattern that is perhaps not so easy/familiar to interpret. Also, what is the overall consensus on the interpretability of suppression effects vs. "normal" mediation effects?

Thank you for this suggestion. We have added two more references (Rucker et al., 2011; Shrout & Bolger, 2002) to literature discussing suppression effects on page 14 (lines 20-21) of the manuscript to show that they are not uncommon within social psychology. Furthermore, we now acknowledge more clearly in the discussion that various interpretations are possible for the suppression effect and that, based on the current data, we cannot draw conclusions what causes the effect.

Concerning the question on the consensus on the interpretability of suppression effect vs. “normal” mediation effects, Rucker et al. (2011) pose that *“Although mediation and suppression effects can be tested using the same statistical method, the key difference between mediation and suppression effect lies in the relationship between the indirect effect (a·b) and the total effect (c). If the indirect effect has the same sign as the total effect, the intervening variable is viewed as a mediator. If an indirect effect has the opposite sign of the total effect, the intervening variable is a suppressor because it weakens the observed relationship by its omission”*. (p. 367).

2.7. Prosociality: This is an intriguing additional finding. You may want to consider briefly contextualizing this with respect to the wider context of research on Moral Typecasting Theory. E.g., there has been some debate in the literature whether harmful agents are granted more agency (e.g., Gray, 2010), or instead rather denied agency (Khamitov et al., 2016; see also Swiderska & Küster, 2020). What are the implications of the current work on these types of more dyadic effects examined by the literature so far? I can see that extending this to a level of general prosociality (also towards non-involved others) is an interesting result as such – yet you might first want to state that prior work has primarily focused just on that dyad (I saw the reference to Jung & Hinds, 2018; yet you may want to expand that towards MTT).

We thank the reviewer for this suggestion. We incorporated this suggestion and the recommended literature in the manuscript (page 6, lines 5-7) as follows:

“So far, studies that investigated the perception of moral agents have focused on how harmful and benevolent behaviour affects perceptions of agency (Gray, 2010; Khamitov et al., 2016; Swiderska & Küster, 2020)”.

2.8. Conclusion: As in the abstract, the conclusion includes a statement that appears to interpret the absence of a clearly significant statistical effect as if it was proving the H0 (p.27 “This effect occurs regardless of the robot’s ability to detect and simulate emotions.”). I believe this would need to be corrected / rephrased, as the paper finds at least some tentative evidence for a (likely) role of emotions across both experiments (see also the comment on the abstract above). Based on your results, it is unclear if emotions play a role or not (with some of your evidence suggesting that it may). I think this should be rephrased – and then the discussion of limitations and potential future work could also consider potential changes to the approach that might yield more conclusive results concerning this hypothesis.

As the reviewer suggested, we have somewhat toned down our statements regarding the lack of evidence for the interaction effect in both the abstract and the discussion. We would like to point out, however, that although the means are in the expected direction, the significance levels of the interaction effects of harm on the perceived capacity for pain were well above the threshold for significance (experiment 1: $p = .67$, experiment 2: $p = .24$). In line with these findings, the Bayesian analyses also found stronger evidence for the model including main effects only, compared to the

model that included the interaction effect. We thus believe that, taken together, we do not find strong support for an interaction effect of harm and emotions.

2.9. Perhaps a revised discussion may also help to better show how this work ties into the mission and aims of *Communications Psychology* as a journal. I would clearly see the potential communicative role of emotions as relevant here.

We thank the reviewer for this suggestion. We believe that the revised manuscript now more clearly reflects how the work ties into the scope of the collection on moral cognition, as well as the journal as a whole. We, for example, integrated the reviewer's suggestion to emphasize the communicative role of emotions, as well as their effects on both mind perception, as well as their effects on the perceived status of the robot at the start of the discussion (page 29, lines 2-7):

"The ability for robots to detect and simulate emotions is deemed quintessential to human-robot communication, as it allows for more natural interactions between robots and humans (Broadbent, 2017; Konijn & Hoorn, 2020). At the same time, however, research suggests that this ability influences both the perception of mind in the robot (Thellman et al., 2022), as well as the perceived moral status of the robot (Darling et al., 2015; Nijssen et al., 2019)."

2.10. The description of the "No Harm" condition of experiment 2 is a bit ambiguous. Is this supposed to be a neutral "no harm" condition, or was this intended as a benevolent/beneficial condition? It is called simply "no harm" in the supplementary materials - yet in the ms it is called "beneficial (no harm condition)" (p.17). This might be a bit confusing and seems somewhat inconsistent. Note also that beneficial agents have been studied in the past as well (e.g., Tanibe et al., 2017; and Küster & Swiderska, 2020) - therefore, you may want to be very clear about what this condition was intended to be.

We thank the reviewer for pointing out this inconsistency. The control condition was intended as a "no harm", rather than a "beneficiary" condition, which we now phrased accordingly in the manuscript.

In our attempt to mirror the harm-condition, in which we emphasized how the moral agent was aware that his actions were harmful, we indeed ended up stating that the moral agent in the "no harm" condition was aware that his actions were beneficial to the robot in the stimulus material for experiment 2. The effect of benevolent intentions towards robots on mind perception have indeed been studied. In the study by Küster and Swiderska (2020), however, no effect of benevolent intentions (vs malevolent intentions) on mind perception was detected. In the study by Tanibe et al. (2017), repairing a robot was found to increase mind perception to the robot through an increase in the perceived capacity to experience joy, though only if the participant took on the perspective of the moral agent.

To rule out that the beneficiary intentions of the moral agent in the no-harm condition in experiment 2 confounded the effect of harm on mind perception, we checked for significant differences in the robot's perceived capacity to experience joy. There was no significant difference between the harm/no-harm condition in the perceived

capacity to experience joy (harm condition: $M = 2.40$, $SD = 1.86$, no harm condition: $M = 2.38$, $SD = 1.92$, $F(1, 673) = .036$, $p = .85$), nor was there a significant interaction effect between harm and emotion on the perceived capacity to experience joy, $F(1, 663) = 2.22$, $p = .125$. Therefore, we conclude that the beneficial intention of the moral agent in the no-harm condition did not confound the harm-made mind effect.

In the revised version of the manuscript, we now elaborate on this further in the discussion (p. 32, lines 10-15):

“...research has also found that caring for a robot by repairing it when it breaks (compared to moving it to an adjacent room and doing nothing) increased mind perception through the perceived capacity to experience joy, though only when the participant took on the perspective of the moral agent and not when taking on the perspective of a bystander (Tanibe et al., 2017). Given that participants in this study were placed in a bystander perspective, it is unlikely that such an effect occurred for our control condition.”

2.11. There were a few instances where I was not sure if the term "victimization" was used correctly. This may require some clarification. I.e., I am more used to seeing this term in the context of victimization of (fully conscious) humans, who are then denied agency.

We use the term “victimization” to refer to the act of being made a victim, as it is the same terminology was used in the original paper by Ward et al. (2013). We have clarified the term after by providing a short explanation after its first use on page 27, line 21-22.

References

Gray, K. (2010). Moral Transformation: Good and Evil Turn the Weak Into the Mighty. *Social Psychological and Personality Science*, 1(3), 253–258. <https://doi.org/10.1177/1948550610367686>

Khamitov, M., Rotman, J. D., & Piazza, J. (2016). Perceiving the agency of harmful agents: A test of dehumanization versus moral typecasting accounts. *Cognition*, 146, 33–47. <https://doi.org/10.1016/j.cognition.2015.09.009>

Swiderska, A., & Küster, D. (2020). Robots as Malevolent Moral Agents: Harmful Behavior Results in Dehumanization, Not Anthropomorphism. *Cognitive Science*, 44(7). <https://doi.org/10.1111/cogs.12872>

Response to reviewer #3

We thank the reviewer for their extensive review, pointing out weaknesses in the argumentation of the previous version of the manuscript, and making suggestions for further improvement. The reviewer raised some concerns regarding the stimulus material and the statistical analyses reported in the paper. We address these concerns below.

3.1. The authors did not directly compare the studies that did vs. did not include the emotion capacity text that Ward used, but instead they compared the Ward version with a new version that explicitly denied emotion capacities. This, to me, is not a control condition but a manipulation of absent emotion capacities. As a result, the (already small) effects we find in the present results don't fully address one of the key motivations of the project: to assess the impact of the Ward et al. decision to equip the robot with emotion capacities. Side note: The authors offer a justification for why they created an emotion-absent condition, but I was not persuaded. They write that not mentioning emotion capacity "would have resulted in a high variety of perceptions." That is the nature of baseline control conditions, which naturally have more error variance that is not controlled for.

As the reviewer noted, we have already included a justification for this design choice in the manuscript. Nevertheless, we agree with the reviewer that this could be extended, and in the revised version, we elaborate more on the manipulation of the emotional capabilities of the robot as a limitation of the textual vignettes in the discussion (page 31, lines 4-19):.

"...more realistic stimulus material like (videos of) actual robots, would have allowed for a more naturalistic manipulation of both simulated emotions as well as intentional harm. Simulated emotions could be manipulated more naturally when using (videos of) actual robots by having the robot either respond emotionally before, during, and after the harm, or not. Furthermore, using (videos of) robots as stimulus material would rule out the possibility that the results are influenced by specific wordings (i.e., stabbing and twisting a scalpel in the robot's sensor vs. kicking the robot's surface). There would also not be a need to specify the moral agent's motivation behind the harm when using a video.

Using textual vignettes as stimulus material, we were not able to find evidence that the harm-made mind effect is moderated by the robot's ability to detect and simulate emotions. However, prior studies did find evidence that mind perception in robots is moderated by (humanlike) characteristics of the robot (Konijn & Hoorn, 2020; Thellman et al., 2022). Additionally, people perceive robots more positively when they are physically present (Li, 2015), and empathize more with physically present robots than mediated ones (Seo et al., 2015). Therefore, it is possible that a more naturalistic manipulation of simulated emotions might still reveal a significant role of simulated emotions in the harm-made mind effect. "

3.2. Though valuable as a replication and extension (testing absence vs. presence of baseline emotion capacities), the manuscript does not offer substantial theoretical advances. Study 1 is primarily motivated by addressing a potential problem in Ward et al., and Study 2 is motivated by addressing a potential problem in Study 1. Thus, I see some failed opportunities:

- To better understand an effect that is quite surprising.
- To improve measurement of the one-shot pain perception variable. For example, one could assess it within-subject: before and after the information about the perpetrator's harm. One could ask for a "first intuition..." after the story setup and then an update "after you know a little more about the robot." A recent paper convincingly argues for such w/s measurements to capture actual cognitive processes:

§ McManus, R. M., Young, L., & Sweetman, J. (2023). Psychology is a property of persons, not averages or distributions: Confronting the group-to-person generalizability problem in experimental psychology. *Advances in Methods and Practices in Psychological Science*, 6(3), <https://doi.org/10.1177/25152459231186615>

We respectfully disagree with the reviewer that our paper missed an opportunity to better understand an effect that is quite surprising: we aimed to test a boundary condition for the effect and found null results. These null results tell us something important about the robustness of the surprising effect.

We very much agree with the reviewer that the measurement of pain perception could be improved. The reviewer might be happy to hear that we have looked at the harm-made mind effect using repeated measurements in ongoing work and found comparable results as reported in this paper.

We have added the improvement of the measurement of pain perception, including the reference that the reviewer suggested, in the discussion section as a suggestion for future research (page 31/32, lines 23-5).

“Our analyses were furthermore conducted solely on a group-level. Research has however shown that there are individual differences in both the tendency to perceive mind in nonhuman agents (Gray et al., 2011; Tharp et al., 2017) as well as in how people respond to robots (Hinz et al., 2019; Spatola, 2021). It has been suggested that, in such cases, group-level findings may not always translate to the person-level (the group-to-person generalizability problem, see McManus et al., 2023). An interesting avenue for future research could therefore be to explore the harm-made mind effect on the person-level through within-subjects designs.”

3.3. There is substantial experimenter demand in the original and the new studies; the authors don't really address this issue. Specifically, I have concerns about the harm manipulation, which the authors took unchanged from Ward et al.:

The act of taking “a scalpel and stab it into George's sensors, twisting the scalpel, thereby damaging George's sensors” highlights sensors and the affected experiences, and it implies that the perpetrator believes his acts will cause some damage to the victim's sensors/experiences – both of which are quite leading pieces of information. A cleaner harm manipulation would be to harm the robot's surface (e.g., by kicking).

Please see our explanation under point 2.1.

3.4 The scalpel stabbing is also likely to cause empathic responses. This empathy, not the act of harming, could be the critical mechanism. A narrative that triggers similar empathic responses without an act of harm could be something like this:

The robot stumbles, and one of its eyes hits a nail sticking out of the floor board; as it tries to free itself, it twists the nail even deeper into its eyes.

The reviewer is correct in assuming that the act of harm triggers empathy for the robot (see, for example, Rosenthal-von der Pütten et al, 2013). However, it seems unlikely for us that empathy, rather than harm, is the critical mechanism that drives mind perception. Firstly, Küster and Swiderska (2020) showed that harm influences both pain perception and empathy, and that empathy and pain perception contribute

to mind perception individually. Secondly, Ward et al. (2013) showed that accidental harm as described by the reviewer does not evoke a harm-made mind effect. Importantly, the intentionality behind the harm is a crucial for the harm-made mind effect to occur.

3.5. The phrase in the control condition, "James will re-oil George's circuits, allowing the robot to update the day's experience into his hardware" is both odd (circuits are not being "oiled") and highlights hardware quite different from the harm condition, which highlights sensors. A better control would be to do maintenance on the robot's sensors.

We acknowledge that there is a slight discrepancy between the control condition and the harm condition concerning the moral agent's actions. However, the most important difference that exist between these two conditions is that the moral agent's actions are perceived as significantly more morally wrong in the harm condition, compared to the no-harm condition. As this was confirmed in the manipulation check, we deem it unlikely that the circuits vs. sensors discrepancy had a significant impact on our findings.

3.6. In their analyses, the authors' correlate the single pain item ("how much the robot is capable of experiencing pain") with the whole mind perception scale. However, this scale contains an item that refers to "experiencing physical or emotional pain." This creates a spurious overlap, a boosted correlation, and may even contribute to the suppressor effect in the mediation analysis (because the boosted mediator-outcome correlation is larger than the IV-mediator correlation)

We apologise for this unclarity in the earlier version of the manuscript. As shown by table S2a, S2b and S3a the supplemental material, the overall mind attribution scale does not include any items referring to "experiencing physical or emotional pain".

Of course, we understand the reviewer's concern regarding the possibility of a boosted mediator-outcome due to spurious overlap in items, and we therefore re-ran our analyses with newly computed variables for overall mind perception and perceived experience that excluded the three overarching items "George can experience feelings", "George can experience emotions", and "George has a personality" (the latter also having the lowest factor loading as shown by table S3a in the supplemental material). Table 1 and Table 2 show the comparison of the results when including/excluding these items.

For experiment 1, the exclusion of these items from the mind perception scale leads to a slight decrease in the coefficient for the direct negative effect of harm on mind perception (from $b = -.18$ to $b = -.16$) and an slight increase in the p-value for the effect (from $p = .0324$ to $p = .0526$). As a result, the p-value for the direct negative effect of harm on mind perception just exceeds the threshold for significance. There are no other significant differences between the scale that includes and the scale that excludes the items.

In Experiment 2, the exclusion of the items has similar consequences for the size of the coefficients but does not affect the significance levels. Given these minimal changes, and to stay in line with what we have preregistered, we decided to retain the analyses with the included items. Should the editor feel that it is more appropriate

to exclude the items and thus divert from the pre-registration, we are of course willing to do so.

Table 1: Comparison of effects on mind attribution when including/excluding overarching items (Experiment 1)

Path	Items included	Items excluded
DV: Mind perception (overall)		
Pain → Mind perception	$b=.42$ $p<.001$	$b=.39$ $p<.001$
Harm → Mind perception (direct effect)	$b=-.18$ $p=.0324$	$b=-.16$ $p=.0526$
Harm → Pain → Mind perception (indirect effect)	$b=.13$ 95%CI[.006, .25]	$b=.12$ 95%CI[.007, .23]
DV: Experience		
Pain → Experience	$b=.52$ $p<.001$	$b=.50$ $p<.001$
Harm → Experience (direct effect)	$b=-.19$ $p<.05$	$b=-.17$ $p<.05$
Harm → Pain → Experience (indirect effect)	$b=.15$ 95%CI[.009, .30]	$b=.15$ 95%CI[.009, .30]

Table 2: Comparison of effects on mind attribution when including/excluding overarching items (Experiment 2)

Path	Items included	Items excluded
DV: Mind attribution (overall)		
Pain → Mind attribution	$b=.49$ $p<.001$	$b=.46$ $p<.001$
Harm → Mind attribution (direct effect)	$b=-.21$ $p<.001$	$b=-.21$ $p<.001$
Harm → Pain → Mind attribution (indirect effect)	$b=.20$ 95%CI[.10, .30]	$b=.19$ 95%CI[.09, .29]
DV: Experience		
Pain → Experience	$b=.60$ $p<.001$	$b=.59$ $p<.001$
Harm → Experience (direct effect)	$b=-.22$ $p<.001$	$b=-.22$ $p<.001$
Harm → Pain → Experience (indirect effect)	$b=.245$ 95%CI[.12, .38]	$b=.24$ 95%CI[.11, .37]

3.7. If the suppressor effect holds after removing spurious overlap, what does the effect tell us? In the authors' discussion of this question (pp. 24-25), I was not convinced by the dehumanization interpretation. Why dehumanize a fictitious robot in order to justify a fictitious person's morally wrong action? And why dehumanize by means of "general" mind perception and not with the more obvious specific perception of pain sensitivity? The multi-dimensional nature of mind perception appears to be a more promising route. The common conception is to think of two dimensions of mind perception: experience and agency, but the authors' Study 1 illustrates an alternative conception, which includes a third dimension (see some references below). It seems more plausible that people respond to the heavy-handed harm manipulation along one dimension (pain/experience) but not at all—or even less so—along other dimensions (agency, social moral capacity)

- o Weisman, K., Dweck, C. S., & Markman, E. M. (2017). Rethinking people's conceptions of mental life. *Proceedings of the National Academy of Sciences of the United States of America*, 114(43), 11374-11379.
- o Malle, B. F. (2019). How many dimensions of mind perception really are there? In E. K. Goel, C. M. Seifert, & C. Freksa (Eds.), *Proceedings of the 41st Annual Meeting of the Cognitive Science Society* (pp. 2268-2274). Cognitive Science Society. READ THESE

The suggestion that people respond differently along the different dimensions of mind perception as an explanation for the suppression effect is something that we have indeed considered in previous versions of the manuscript, which also contained additional mediation analyses for the separate dimensions of mind perception (as detected by our principal component analyses). In the revised version of the manuscript, we have now included these analyses again (Experiment 1: page 15, lines 4-16 + Figure 2 on page 16; Experiment 2: Page 24, lines 9-21 + Figure 4 on page 25).

Importantly, in our opinion, these analyses provide further support for our dehumanization account. The analyses show that 1) there is a significant indirect effect of harm on each dimension of mind perception through the perceived capacity for pain and 2) that the suppression effect seems to occur only for the experience dimension. Thus, there seems to be a *mechanistic dehumanization* effect. Mechanistic dehumanization rests in the direct contrast of humans with machines, which includes a denial of human nature and emotions (Haslam, 2006) To further solidify our dehumanization account, we have added literature on possible motivations people could have to dehumanize a (harmed) robot, for example, because of the uncanny valley of mind (Stein & Ohler, 2017).

3.8. As the authors point out, "participants generally did not believe the robot was capable of experiencing pain." This raises the question of what the effect actually consists of: What sort of "mind" is "made by harm"? How many people are moved at all by the harm?

We agree with the reviewer that this information was lacking from the previous version of the manuscript. We have now added tables that display the descriptive

variables for pain, overall mind perception, and agency and experience, respectively. These can be found on pages 14 and 23 of the revised manuscript.

3.9. I found the perceived prosociality questions to be distracting; they have really very little to do with the main questions at hand.

The perceived prosociality questions are indeed not the main focus of the manuscript. To keep in line with our preregistration, and given the fact that the other reviewers did find the analyses of interest, we decided to retain the prosociality questions in the manuscript. We did, however, elaborate further on the inclusion of these hypotheses in the introduction of the revised manuscript (page 6, lines 5-10):

“So far, studies that investigated the perception of moral agents have focused on how harmful and benevolent behaviour affects perceptions of agency (Gray, 2010; Khamitov et al., 2016; Swiderska & Küster, 2020) One study furthermore reported that watching a video of someone “torturing” a robotic dinosaur caused feelings of anger towards the “torturer” amongst viewers (Rosenthal-von der Pütten et al., 2013). The question if moral transgressions against a robot leads to judgements of character so far remained unanswered.”

We also emphasized on page 5, lines 20-21 that the main focus of the paper is to test if robots that simulate emotions evoke a stronger harm-made mind effect when they can detect and simulate emotions.

3.10. The extremely skewed pain perception variable may be best analyzed as a dichotomous “no pain” vs. “some pain” variable. An ANOVA is simply not appropriate for such a distribution. That would also put an intuitive effect size on the harm effect: How many people in each condition attribute at least some pain to the robot?

Analysing the pain perception variable as a dichotomous variable would not allow for a mediation analyses. Fortunately, the ANOVA is quite robust to violations of the assumption of normality, as the data would have to be normally distributed within the population, not the sample. The mediation analyses uses bootstrapping, so the problem of the skewness should also be reduced in these analyses.

3.11. What did participants’ verbal comments reveal? Did they comment on any of the strange aspects of the study

We provided participants an opportunity to leave comments in an open text field using the prompt “If you have any thoughts or comments about the study you would like to share, you can leave them here”. These comments therefore include substantive as well as non-substantive remarks (e.g., “Interesting study, thank you”).

Several participants in the no-harm conditions commented that they had difficulty answering the questions regarding the prosociality of the moral agent (e.g., “I didn’t feel there was enough information about James to make any judgement on his character”). Some participants in the harm conditions indicated that they felt strongly about James’ actions (e.g., “Someone should certify James as having mental health issues”).

Some participants expressed empathy for the robot (e.g., “Poor George”, “George deserved better”) and also elaborated on their responses to the stimulus material (e.g., “I had a visceral reaction upon reading about him stabbing the scalpel, logically I don’t think there’s been any harm done but emotionally I do.”)

And, indeed, as quite common in research on robot abuse, some participants commented on the study's strangeness (e.g., "Weird but funny", "Slightly strange survey. Emotions of a robot?").

3.12. The statistical test of two main effects and an interaction (Harm x Emotion) does not seem to be in keeping with the authors' original observation that emotion capacity might facilitate the Harm effect. An interaction effect has specific contrast weights: +1, -1, -1,+1 (in the four cells of the design), which means that the F test for the interaction assesses the fit of a complete reversal of the pattern in the no emotion condition (namely, more pain for no-harm than for harm), controlling for main effects. The most appropriate test to examine the facilitating emotion capacity information would be two simple main effects: one that looks at the "Ward case" (with emotion capacity mentioned), the other looks at the "no-emotion capacity" case. Does the effect hold only in the first? How strong is the effect in each case? If the effect is not significant in the second, does the Bayesian analysis favor a true null finding?

We are not sure we fully understand the reviewer's remark concerning the interaction effect. We consulted several statistical experts on this matter, and to their and our knowledge, the ANOVA tests both for ordinal (parallel) as well as disordinal (non-parallel) interactions. Furthermore, our samples were not powered for the analysis that the reviewer proposes, which would require the dataset to be split into two separate sets. Conducting these analyses would therefore result in a lack of power which would not allow us to draw confident conclusions about (the absence of) effects.

Furthermore, the Bayesian statistics for all ANOVA-analyses were included in the supplemental material, which showed stronger evidence for a model including only main effects over the null model, compared to a model including main and interaction effects over the null model. It therefore seems fitting to us to keep in line with our preregistered analysis.

3.13. I'd like to see the full model of the mediation, including Emotion capacity, like it is shown in the Supplementary material for "Consciousness." Alternatively (and as I suggested above for the ANOVA), two separate model paths could be assessed for the No-Emotion condition and Yes-Emotion condition, to get clear on the impact of the potential facilitator in Ward et al.

In the new version of the manuscript, we have included the full models of the mediation for both the complete mind attribution scale, as well as the separate subscales for agency and experience (see page 16 and 25 of revised manuscript).

3.14. How was "Consciousness" measured? One item is already included in the standard Gray & Wegner scale, phrased as "having experiences and being aware of things." Did the authors use a separate scale or item or Ward et al.'s two items? What do the authors take away from the additional analysis?

In the new version of the manuscript, we explain how we measured consciousness in the footnotes on pages 11 and 21. The results for consciousness largely mirror those

for mind perception: harm as a significant indirect effect on perceived consciousness of the robot through the perceived capacity for pain, but there is no direct relationship between harm and perceived consciousness. Emotions have a positively affect the perceived consciousness of the robot both directly and indirectly. The results for consciousness can be found in Figure S1 of the supplemental material.

3.15. “we are fairly confident that our results are generalizable” – to what populations, methods, cultures, etc.?

We have now specified that the samples were taken from western cultures.

References

- Gray, K. (2010). Moral Transformation: Good and Evil Turn the Weak Into the Mighty. *Social Psychological and Personality Science*, 1(3), 253–258. <https://doi.org/10.1177/1948550610367686>
- Gray, K., Jenkins, A. C., Heberlein, A. S., & Wegner, D. M. (2011). Distortions of mind perception in psychopathology. *Proceedings of the National Academy of Sciences*, 108(2), 477–479. <https://doi.org/10.1073/pnas.1015493108>
- Haslam, N. (2006). Dehumanization: An Integrative Review. *Personality and Social Psychology Review*, 10(3), 252–264. https://doi.org/10.1207/s15327957pspr1003_4
- Hinz, N.-A., Ciardo, F., & Wykowska, A. (2019). Individual Differences in Attitude Toward Robots Predict Behavior in Human-Robot Interaction. In M. A. Salichs, S. S. Ge, E. I. Barakova, J.-J. Cabibihan, A. R. Wagner, Á. Castro-González, & H. He (Eds.), *Social Robotics* (Vol. 11876, pp. 64–73). Springer International Publishing. https://doi.org/10.1007/978-3-030-35888-4_7
- Khamitov, M., Rotman, J. D., & Piazza, J. (2016). Perceiving the agency of harmful agents: A test of dehumanization versus moral typecasting accounts. *Cognition*, 146, 33–47. <https://doi.org/10.1016/j.cognition.2015.09.009>
- Konijn, E. A., & Hoorn, J. F. (2020). Differential Facial Articulatory in Robots and Humans Elicit Different Levels of Responsiveness, Empathy, and Projected Feelings. *Robotics*, 9(4), 92. <https://doi.org/10.3390/robotics9040092>
- Küster, D., & Swiderska, A. (2021). Seeing the mind of robots: Harm augments mind perception but benevolent intentions reduce dehumanisation of artificial entities in visual vignettes. *International Journal of Psychology*, 56(3), 454–465. <https://doi.org/10.1002/ijop.12715>
- Li, J. (2015). The benefit of being physically present: A survey of experimental works comparing copresent robots, telepresent robots and virtual agents. *International Journal of Human-Computer Studies*, 77, 23–37. <https://doi.org/10.1016/j.ijhcs.2015.01.001>
- McManus, R. M., Young, L., & Sweetman, J. (2023). Psychology Is a Property of Persons, Not Averages or Distributions: Confronting the Group-to-Person Generalizability Problem in Experimental Psychology. *Advances in Methods and Practices in Psychological Science*, 6(3), 25152459231186615. <https://doi.org/10.1177/25152459231186615>
- Rosenthal-von der Pütten, A. M., Krämer, N. C., Hoffmann, L., Sobieraj, S., & Eimler, S. C. (2013). An Experimental Study on Emotional Reactions Towards a Robot. *International Journal of Social Robotics*, 5(1), 17–34. <https://doi.org/10.1007/s12369-012-0173-8>

- Seo, S. H., Geiskkovitch, D., Nakane, M., King, C., & Young, J. E. (2015). Poor Thing! Would You Feel Sorry for a Simulated Robot?: A comparison of empathy toward a physical and a simulated robot. *Proceedings of the Tenth Annual ACM/IEEE International Conference on Human-Robot Interaction*, 125–132. <https://doi.org/10.1145/2696454.2696471>
- Spatola, N. (2021). The personality of anthropomorphism: How the need for cognition and the need for closure define attitudes and anthropomorphic attributions toward robots. *Computers in Human Behavior*, 122, 106841. <https://doi.org/10.1016/j.chb.2021.106841>
- Stein, J.-P., & Ohler, P. (2017). Venturing into the uncanny valley of mind—The influence of mind attribution on the acceptance of human-like characters in a virtual reality setting. *Cognition*, 160, 43–50. <https://doi.org/10.1016/j.cognition.2016.12.010>
- Swiderska, A., & Küster, D. (2018). Avatars in Pain: Visible Harm Enhances Mind Perception in Humans and Robots. *Perception*, 47(12), 1139–1152. <https://doi.org/10.1177/0301006618809919>
- Swiderska, A., & Küster, D. (2020). Robots as Malevolent Moral Agents: Harmful Behavior Results in Dehumanization, Not Anthropomorphism. *Cognitive Science*, 44(7), e12872. <https://doi.org/10.1111/cogs.12872>
- Tanibe, T., Hashimoto, T., & Karasawa, K. (2017). We perceive a mind in a robot when we help it. *PLOS ONE*, 12(7), e0180952. <https://doi.org/10.1371/journal.pone.0180952>
- Tharp, M., Holtzman, N. S., & Eadeh, F. R. (2017). Mind Perception and Individual Differences: A Replication and Extension. *Basic and Applied Social Psychology*, 39(1), 68–73. <https://doi.org/10.1080/01973533.2016.1256287>
- Thellman, S., De Graaf, M., & Ziemke, T. (2022). Mental State Attribution to Robots: A Systematic Review of Conceptions, Methods, and Findings. *ACM Transactions on Human-Robot Interaction*, 11(4), 1–51. <https://doi.org/10.1145/3526112>
- Yam, K. C., Bigman, Y., & Gray, K. (2021). Reducing the uncanny valley by dehumanizing humanoid robots. *Computers in Human Behavior*, 125, 106945. <https://doi.org/10.1016/j.chb.2021.106945>

23rd Apr 24

Dear Ms Wieringa,

Your manuscript titled "Replicating and Extending the Harm-Made Mind Effect in Robots: The Role of Emotions" has now been seen by our reviewers, whose comments appear below. In light of their advice I am delighted to say that we are happy, in principle, to publish a suitably revised version in *Communications Psychology* under the open access CC BY license (Creative Commons Attribution v4.0 International License).

We therefore invite you to revise your paper one last time to address the remaining concerns of our reviewers and a list of editorial requests. At the same time we ask that you edit your manuscript to comply with our format requirements and to maximise the accessibility and therefore the impact of your work.

EDITORIAL REQUESTS:

SUBMISSION INFORMATION:

OPEN ACCESS:

Communications Psychology is a fully open access journal. Articles are made freely accessible on publication under a CC BY license (Creative Commons Attribution 4.0 International License). This

license allows maximum dissemination and re-use of open access materials and is preferred by many research funding bodies.

For further information about article processing charges, open access funding, and advice and support from Nature Research, please visit <https://www.nature.com/commspsychol/article-processing-charges>

At acceptance, you will be provided with instructions for completing this CC BY license on behalf of all authors. This grants us the necessary permissions to publish your paper. Additionally, you will be asked to declare that all required third party permissions have been obtained, and to provide billing information in order to pay the article-processing charge (APC).

* **DATA AVAILABILITY:**

[link redacted]

Best regards,

Jennifer Bellingtier

Jennifer Bellingtier, PhD

Senior Editor

Communications Psychology

Yafeng Pan, PhD

Editorial Board Member

Communications Psychology

orcid.org/0000-0002-5633-8313

REVIEWERS' EXPERTISE:

Reviewer #1: Harm-made mind/Dyadic morality

Reviewer #2: Harm-made mind/Online survey

Reviewer #3: Harm-made mind/Dyadic morality/Online survey

REVIEWERS' COMMENTS:

Reviewer #1 (Remarks to the Author):

The authors did an excellent job addressing my concerns, and I am pleased with the overall manuscript. Nice work!

Reviewer #2 (Remarks to the Author):

I would like to thank the authors for their hard work in thoroughly and transparently revising their manuscript!

Upon re-reading the revised version and rebuttal letter, I found all of my previously raised suggestions to have been sufficiently addressed. I particularly appreciated the additional work that has gone into the supporting analyses to address some of the points, such as whether the emotion-inducing scalpel imagery in the harm condition might have decreased the effectiveness of the emotion condition itself. On this point, I think that there is still the possibility, that a stronger/cleaner difference between emotion-inducing (Emotion) and not-emotion-inducing (No Emotion) condition might provide a better basis for testing for the hypothesized interaction effect. However, I believe this point has been addressed as well as it could be given the available data.

The added Bayesian analyses in response to the comments by reviewer #3 also appear to underscore the methodological soundness of this approach.

Reviewer #3 (please see attachment)

I appreciate the authors' careful responses to all the reviewers' comments. I continue to be concerned about the psychological impact of the original manipulation by Ward et al., but that is not the present authors' fault. I wish, however, they would comment on this manipulation in the General Discussion (another reviewer expressed related, though not identical concerns).

The authors and I disagree about the theoretical value of a null finding. They “aimed to test a boundary condition for the effect and found null results. These null results tell us something important about the robustness of the surprising effect.” This null effect may show robustness but does not reduce our surprise as it does not explain any causal factor in bringing about the effect. There are many reasons why the manipulation didn't work (as other reviewers pointed out, it could have been drowned out by the harm manipulation; it could be that the claimed absence of emotion capacities paired with striking reference to sensors etc. made participants disbelieve the “absence”, etc.). We just don't know.

But there is actually something more interesting here that I hadn't fully recognized before, until I saw the newly added Table 2. It is the fact that the basic Ward et al. main effect of harm on mind perception did not replicate at all. Below is the relevant portion of Table 2 (Emotion condition) to directly compare to the results from Ward et al.'s Study 3 below.

	Emotions			
	Harm		No Harm	
	M	SD	M	SD
Pain	2.95	1.54	2.61	1.66
Mind	3.76	1.17	3.82	1.11
Agency	4.13	1.42	4.37	1.27
Experience	3.55	1.34	3.61	1.29

“Participants in the harm condition, relative to those in the control condition, attributed to the robot a higher capacity to experience pain (control condition: $M = 3.00$, $SD = 1.29$; harm condition: $M = 4.63$, $SD = 1.02$), $t(119) = 4.84$, $p < .0001$, $d = 0.89$, as well as **more mind overall** ($\alpha = .94$; control condition: $M = 3.79$, $SD = 1.33$; harm condition: $M = 4.33$, $SD = 1.26$), $t(119) = 2.31$, $p = .02$, $d = 0.42$; see Figure 1.”

I think this deserves more attention in the manuscript. The mediation analyses in Ward et al. were motivated by the desire to better understand an *effect* of one variable (harm) on another (mind perception) by way of a third variable (pain perception). But there is no such *effect* here of harm on mind perception. Ward et al. always performed tests of this basic effect first, and I think the present authors should do so as well and acknowledge that it doesn't replicate. I don't object to running a mediation *in addition* (though it should also be made clear that many authors¹ have expressed concerns about mediation analyses in which the mediator is not manipulated). In a way, the suppressor effect now becomes more interesting, because the present data suggest that we shouldn't talk about “harm-made mind” but about two separate effects of observed harm inflicted on robots: one increases perceived pain capacity and thereby perceived mind capacities more generally, and the other decreases perceived mind capacities, perhaps as a form of mechanization (I suggest not using the word de-humanization for responses to robots).

¹ For a recent formulation: Bullock, J. G., & Green, D. P. (2021). The failings of conventional mediation analysis and a design-based alternative. *Advances in Methods and Practices in Psychological Science*, 4(4), 25152459211047227. Also of relevance: Fiedler, K., Schott, M., & Meiser, T. (2011). What mediation analysis can (not) do. *Journal of Experimental Social Psychology*, 47(6), 1231–1236.

Reviewer #1 (Remarks to the Author):

The authors did an excellent job addressing my concerns, and I am pleased with the overall manuscript. Nice work!

We thank the reviewer for taking the time to read through our revised manuscript and are happy to hear that the reviewer is pleased with our revisions.

Reviewer #2 (Remarks to the Author):

I would like to thank the authors for their hard work in thoroughly and transparently revising their manuscript!

Upon re-reading the revised version and rebuttal letter, I found all of my previously raised suggestions to have been sufficiently addressed. I particularly appreciated the additional work that has gone into the supporting analyses to address some of the points, such as whether the emotion-inducing scalpel imagery in the harm condition might have decreased the effectiveness of the emotion condition itself. On this point, I think that there is still the possibility, that a stronger/cleaner difference between emotion-inducing (Emotion) and not-emotion-inducing (No Emotion) condition might provide a better basis for testing for the hypothesized interaction effect. However, I believe this point has been addressed as well as it could be given the available data.

The added Bayesian analyses in response to the comments by reviewer #3 also appear to underscore the methodological soundness of this approach.

We thank the reviewer for their time and helpful input. We are happy to hear that the reviewer appreciates the additional analyses and is happy with the revised manuscript.

Reviewer #3

I appreciate the authors' careful responses to all the reviewers' comments. I continue to be concerned about the psychological impact of the original manipulation by Ward et al., but that is not the present authors' fault. I wish, however, they would comment on this manipulation in the General Discussion (another reviewer expressed related, though not identical concerns).

The authors and I disagree about the theoretical value of a null finding. They "aimed to test a boundary condition for the effect and found null results. These null results tell us something important about the robustness of the surprising effect." This null effect may show robustness but does not reduce our surprise as it does not explain any causal factor in bringing about the effect. There are many reasons why the manipulation didn't work (as other reviewers pointed out, it could have been drowned out by the harm manipulation; it could be that the claimed absence of emotion

capacities paired with striking reference to sensors etc. made participants disbelieve the "absence", etc.). We just don't know.

We thank the reviewer for their comments. The reviewer is indeed correct that we respectfully disagree with each other concerning the theoretical value of our null findings. We also agree with the reviewer that, with different stimulus material, an interaction between harm and emotions could still be possible. We have taken up the reviewers suggestion to elaborate a little more on the limitations of the harm manipulation in the discussion:

"One could argue, for example, that especially the mention of damaged sensors in the harm conditions of the current stimulus material might have interfered with the emotion manipulation, thereby possibly causing the absence of an interaction effect." (p.23, lines 18-20)

But there is actually something more interesting here that I hadn't fully recognized before, until I saw the newly added Table 2. It is the fact that the basic Ward et al. main effect of harm on mind perception did not replicate at all. Below is the relevant portion of Table 2 (Emotion condition) to directly compare to the results from Ward et al.'s Study 3 below. I think this deserves more attention in the manuscript. The mediation analyses in Ward et al. were motivated by the desire to better understand an *effect* of one variable (harm) on another (mind perception) by way of a third variable (pain perception). But there is no such *effect* here of harm on mind perception. Ward et al. always performed tests of this basic effect first, and I think the present authors should do so as well and acknowledge that it doesn't replicate. I don't object to running a mediation *in addition* (though it should also be made clear that many authors¹ have expressed concerns about mediation analyses in which the mediator is not manipulated). In a way, the suppressor effect now becomes more interesting, because the present data suggest that we shouldn't talk about "harm-made mind" but about two separate effects of observed harm inflicted on robots: one increases perceived pain capacity and thereby perceived mind capacities more generally, and the other decreases perceived mind capacities, perhaps as a form of mechanization (I suggest not using the word de-humanization for responses to robots).

We agree with the reviewer that the absence of the direct effect of Harm on Mind Perception reported in Ward et al. (2013) is surprising and perhaps was not yet highlighted enough in within the manuscript. We now place a clearer emphasis on the absence of that effect in the manuscript:

"In contrast to Ward et al. (2013)¹, who reported a significant positive total effect of Harm on Mind perception, we did not find statistically significant evidence for a total effect of Harm on Mind perception (the c path) in either of our experiments" (Results - p. 16, lines 15-17).

"In both experiments, we found a positive indirect effect of harm on mind perception (the harm-made mind effect), as well as a direct negative effect of harm on mind perception in the

robot. This is in contrast to Ward et al. (2013)¹, who reported a positive direct effect of harm on mind perception in a robot.” (Discussion – p. 20, lines 17-18).